# RAID: Towards Robust AI-Generated Image Detection with Bit Reversed Images

## Abstract

The rapid advancement of image generation models has made it increasingly difficult for people to distinguish AI-generated images from real ones. To prevent the potential risks associated with the misuse of fake images, AI-generated image detection has gained significant attention. Existing methods neglect the inherent differences between real and fake images, thus lacking robustness and generalization ability. In this work, we innovatively investigate AI-generated image detection using bit-planes, and introduce the bit reversed image. We propose a simple yet effective pipeline consisting of construction of bit reversed images, gradient-based patch selection and a convolutional classifier. Extensive experiments on more than 40 benchmarks verify the effectiveness of our approach across different settings, including evaluations of generalization capability and zero-shot performance. Particularly, our approach achieves nearly 100% accuracy on eight benchmarks for cross-generator evaluation on the GenImage dataset.

## 1 Introduction

The realism of images produced by advanced generative models, such as Generative Adversarial Networks (GANs) (Goodfellow et al., 2014) and Diffusion Models (Rombach et al., 2022), has improved dramatically in recent years. This progress raises serious concerns about the potential misuse of AI-generated images (Juefei-Xu et al., 2022), such as the creation of deceptive or harmful content. Such risks underscore the pressing need for robust methods that can accurately differentiate AI-generated images from real ones.

Existing deepfake detection works can be categorized into three groups: spatial domain-based, frequency domain-based, and patch-based approaches. The first category analyzes pixel-level texture patterns and gradient artifacts (Tan et al., 2023) and reconstruction error (Wang et al., 2023b). The second reveals artifacts often imperceptible in pixel space by focusing on frequency artifacts (Zhang et al., 2019) and high-frequency features (Dzanic et al., 2020). The third approach learns and aggregates features from local patches rather than processing the entire image (Chen et al., 2024b; Zheng et al., 2024; Yang et al., 2025). However, these methods employ sophisticated models to learn effective features from images, thereby neglecting to discover the inherent differences between real and fake images.

A grayscale image can be reversibly decomposed into eight bit-planes. Least Significant Bit (LSB) substitution is a well-known technique in the fields of information hiding (Chan & Cheng, 2004) and steganography (Elharrouss et al., 2020). Bit-plane has shown potential and advantages in image stabilization (Ko et al., 1998), image encryption (Gan et al., 2019), image compression (Zhang et al., 2024b), and implicit neural representations (Han et al., 2025). Such bit-plane-based methods are primarily designed for image processing. Since bit-planes inherently possess the ability to convey fine details within an image, they hold the potential to discern subtle differences between real and AI-generated images. However, marrying bit-planes and AI-generated image detection, which is a promising direction, has not been studied yet. In prior research on frequency-based method (Zhong et al., 2023; Tan et al., 2024a), operations such as generator upsampling and regularization losses introduce noticeable discrepancies (artifacts) in the frequency domain between generated and real images. These artifacts manifest as unnatural directional properties in the spectrum of synthetic images, with diffusion-generated images often exhibiting energy concentrated in specific frequency bands that deviate from the uniform attenuation model characteristic of natural images. The signif-

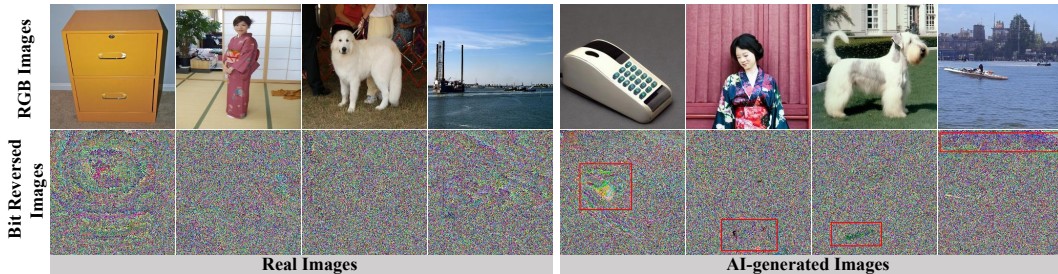

Figure 1: **Comparison of bit reversed images between real and fake images.** We find that noticeable artifacts appear in certain regions of reversed images in fake images. Conversely, the noise in bit reversed images of real images tends to be more naturally distributed.

icant discrepancies between generated and real images across bit-planes particularly in low-order planes provide a strong basis for bit-plane-based Deepfake detection.

To enhance these discrepancies, we innovatively introduce the bit reversed image, which simply reverses the permutation order of bit-planes before constructing the image. The bit reversed image has two distinct characteristics. First, it can be reversibly converted into bit-plane images or the original image. Second, the visual content of the original image is encrypted, while the noise and fine details are amplified. Figure 1 shows the comparison of corresponding bit reversed images between real and AI-generated images. It can be seen that, for fake images, artifacts are apparent in the corresponding bit reversed images.

To this end, we propose a simple yet effective approach for AI-generated image detection. We first synthesize the bit reversed image from the original image based on bit-planes, and then design a patch-based classifier to detect fake content. Specifically, we investigate both bit forward image and bit reversed image during the bit-plane-based image construction. The patch-based classifier consists of the gradient-based patch selection followed by a convolutional classifier. Both construction of bit reversed images and gradient-based patch selection operate at millisecond-level speed and involve no trainable parameters. The convolutional network is adapted to prevent premature feature compression and accommodate small image patches as input. We assess our approach across various AI-generated image detection settings, including cross-generator evaluation, zero-shot generalization, and cross-dataset evaluation. Our approach achieves state-of-the-art performance on more than 40 benchmarks, significantly surpassing existing approaches. In summary, our main contributions are as follows:

- **Innovative deepfake representation:** We innovatively tackle AI-generated image detection based on bit-planes, and introduce the bit reversed image that can be reversibly constructed from the original image.
- **Efficient pipeline design:** We propose a simple yet effective pipeline for AI-generated image detection, named RAID, which significantly outperforms existing approaches on standard benchmarks while operating at the millisecond level.

## 2 RELATED WORK

**AI-Generated Image Detection:** Existing methods can be roughly categorized into spatial domain-based, frequency domain-based, and patch-based approaches. In contrast, this work is among the first bit-plane-based approaches.

For spatial domain-based methods, GLFF (Ju et al., 2023) utilizes a fusion of global and local features to effectively capture inconsistencies at multiple scales. DIRE (Wang et al., 2023b) leverages reconstruction error as a fundamental detector for AI-generated images. LaRE[2] (Luo et al., 2024) incorporates refinement mechanisms for both spatial and channel features to boost feature learning. Jia et al. (2025) exploit color distribution inconsistencies via quantization–restoration analysis.

For frequency domain-based methods, Corvi et al. (2023b;a) extend spectral analysis to diffusion models by identifying unique frequency fingerprints. PatchCraft (Zhong et al., 2023) postulates

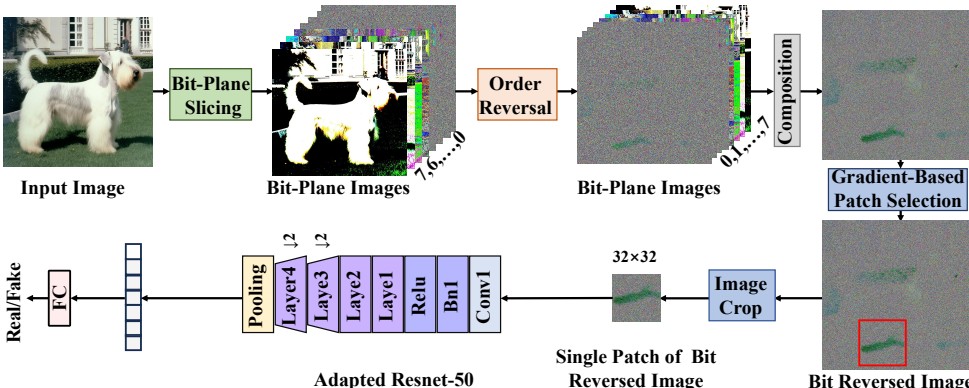

Figure 2: **Pipeline for the proposed method.** Given an RGB image, our approach first extracts the bit reversed image based on bit-planes, then heuristically selects an image patch, and finally performs fake classification using a trainable convolutional network. In the adapted ResNet-50, $\downarrow 2$ indicates a 2× downsampling of spatial resolution in feature learning.

that artifacts predominantly manifest in high-frequency texture regions. FreqNet (Tan et al., 2024a) compels the model to learn source-agnostic features by incorporating high-frequency representation modules and frequency convolution layers into the CNN classifier. Karageorgiou et al. (2025) introduce a method based on frequency reconstruction and reconstruction similarity.

As for patch-based methods, PatchCraft (Zhong et al., 2023) regards the difference between the patches with the highest and lowest diversity as the detection criterion, while SSP (Chen et al., 2024b) selects the patch with the highest diversity to expose artifacts. Other studies challenge the adequacy and effectiveness of relying solely on a single patch or several patches. Zheng et al. (2024) present a classifier trained on patch-shuffled images and aggregates patch-wise features. Yang et al. (2025) randomly replace partial patches with real patches to force the model to learn artifacts from all patches. Xiao et al. (2025) show that high-quality AI-generated image detection can be improved by selecting patches identified through low-level visual cues.

**Bit-Plane-Based Image Processing:** Bit-plane-based operations play a significant role in image processing. A notable example is Least Significant Bit (LSB) substitution (Chan & Cheng, 2004), which is a simple and widely adopted data hiding method. Bit-plane-based methods can be used for reversible data hiding in encrypted images by exploiting intra- and inter-bit-plane correlations or using asymmetric coding (Kumar et al., 2023; Zhang et al., 2024a). They can also be applied to image stabilization (Ko et al., 1998) and image encryption (Gan et al., 2019). Recently, Punnappurath & Brown (2021) propose a bit-plane-wise deep learning framework for bit-depth reconstruction that progressively recovers residuals at each bit-plane level. Zhang et al. (2024b) present bit plane slicing with a dimension-tailored autoregressive model to enhance latent variable use and improve lossless image compression efficiency. Han et al. (2025) introduce a bit-plane decomposition approach for implicit neural representations, enabling faster convergence and lossless fitting of high bit-depth signals. Recently, LOTA (Wang et al., 2025) introduces a bit-planes guided noisy image generation for AI-generated image detection. Different from these works, and similar to LOTA, we study a bit-plane–based method for detecting AI-generated images.

## 3 METHODOLOGY

We address AI-generated image detection from the perspective of bit-planes. The pipeline of our approach is shown in Figure 2. Details are described below.

### 3.1 CONSTRUCTION OF BIT REVERSED IMAGES

A grayscale image can be losslessly decomposed into eight binary images using bit-plane decomposition, each representing a specific bit-plane. For the RGB image, each channel corresponds to a gray-scale image, which can be subsequently decomposed as 8 bit-planes. Higher bit-planes cap-

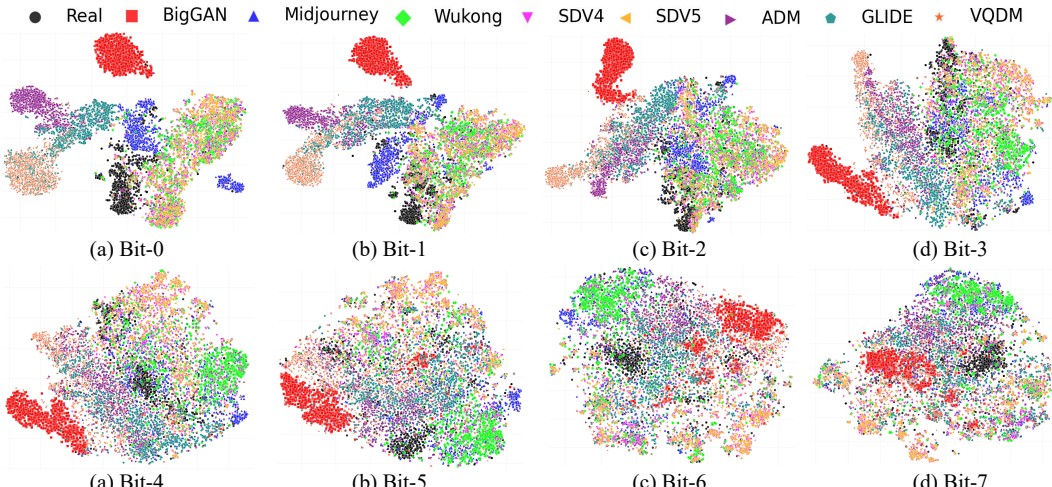

Figure 3: **Comparison of sample distributions between real images and AI-generated images of different bit-planes.** We find that for the low bit-planes, especially bit-0, bit-1, and bit-2, the distributions of real images and AI-generated images are clearly separable, and different types of generators also exhibit distinct sample distributions. In contrast, for the high bit-planes, the distributions of real images and AI-generated images of some generators heavily overlap.

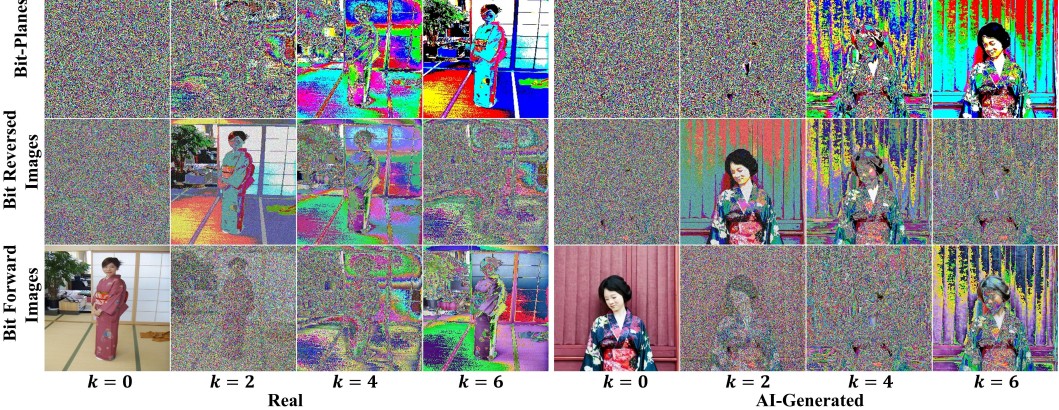

Figure 4: **Visualizations of bit-planes, bit reversed images and bit forward images for real and AI-generated RGB images**. $k = \{0, 1, \ldots, 7\}$ is index of bit-planes. For bit reversed or forward images, $k$ denotes the positions of the left circular shift operation, as defined in Eqs. (4) and (5). For AI-generated images, different values of $k$ result in varying degrees of artifacts.

ture the major structural or low-frequency information, similar to high-frequency components, while lower bit-planes contain finer details, analogous to low-frequency components.

In prior research (Zhong et al., 2023; Tan et al., 2024a) on frequency-based Deepfake detection, operations such as generator upsampling and regularization introduce noticeable discrepancies (artifacts) in the frequency domain between generated and real images. Instead of focusing on high-frequency components, we unveil AI-generated artifacts from the perspective of bit-planes.

Although AI-generated images from current generators look almost indistinguishable from real images, significant discrepancies exist between AI-generated and real images in the low bit-planes, as illustrated in Figure 3. As AI-generated images lack true physical sensor noise, they show structured or unnatural randomness in low-bit planes. Since the neural networks learn semantic correlations during image synthesis, semantic structures may leak into low bit-planes of AI-generated images. In addition, AI-generated images often lack true high-frequency content. Since generative models frequently apply denoising or upsampling, the lower bit-planes of AI-generated images often be-

come correlated with the higher bit-planes. This evidence can be observed in Figure 1 and is also confirmed in LOTA (Wang et al., 2025).

To highlight invisible artifacts in low bit-planes of the AI-generated image, a straightforward idea is to reverse the order of eight bit-planes before recomposing the image. Let $x^c$ denote the $c$-th channel of the RGB image, where $c \in \{R, G, B\}$, and the eight bit-planes are given by $\{x^c_k \mid k = 0, 1, \ldots, 7\}$. The composed image is computed as:

$$\tilde{x}^c = \sum_{k=0}^{7} w_k \cdot x^c_k, \tag{1}$$

where $w_k$ denotes the weight of the $k$-th bit-plane. The eight weights can be written as a weight vector $w = [w_0, w_1, w_2, w_3, w_4, w_5, w_6, w_7]$.

For constructed images, varying the value of $w$ yields composed images of different styles. These images can be categorized into two types: Bit Forward Images and Bit Reversed Images, as visualized in Figure 4.

**Bit Forward Images:** The default weight vector is:

$$w = [2^0, 2^1, 2^2, 2^3, 2^4, 2^5, 2^6, 2^7]. \tag{2}$$

The weights are determined by the positions of the bit-planes. Specifically, for the $k$-th bit-plane, the corresponding weight is $2^k$. The original image can be equivalently recovered using these weights.

Given a weight vector with eight elements, performing a left circular shift by one position iteratively yields eight distinct vectors. The transformed weight vector after performing a left circular shift by $k$ positions is:

$$w = [2^k, 2^{k+1}, \ldots, 2^7, 2^0, \ldots, 2^{k-1}]. \tag{3}$$

**Bit Reversed Images:** The bit reversed image can be obtained by reversing the order bit-planes, and the corresponding weight vector is:

$$w = [2^7, 2^6, 2^5, 2^4, 2^3, 2^2, 2^1, 2^0]. \tag{4}$$

For the $k$-th bit-plane, the corresponding weight is $2^{7-k}$. Since higher bit-planes have smaller weights, the bit reversed image can amplify fine details and noise in the original image.

Similarly, a left circular shift can be applied to generate eight different weight vectors. The resulting vector after performing a left circular shift by $k$ positions is:

$$w = [2^{7-k}, 2^{6-k}, \ldots, 2^0, 2^7, \ldots, 2^{8-k}]. \tag{5}$$

### 3.2 PATCH-BASED CLASSIFIER

After constructing bit-reversed images, we design a patch-based classifier that consists of Gradient-Based Patch Selection and a Convolutional Classifier.

**Gradient-Based Patch Selection:** Although artifacts in bit-reversed images serve as a critical feature for distinguishing real and generated images, they still contain a lot of irrelevant information that may interfere with detection. To mitigate such interference and amplify the artifacts, we introduce Gradient-Based Patch Selection (GBPS) to select the most informative patch from each image.

Given the bit-reversed image $\tilde{x}^c$, we partition it randomly into non-overlapping patches. To evaluate the sparsity of image gradients along various directions, we propose a divergence-based scoring function. For a noisy patch $\tilde{z}_p$, where $p$ represents the patch index, the score $g_p$ is calculated as follows:

$$g_p = \sum_{d \in \mathcal{D}} \left\| \tilde{x}^c * g_d \right\|_1, \tag{6}$$

where $*$ denotes the image convolution operation, $\| \cdot \|_1$ represents the $L1$ norm of the matrix and $\mathcal{D} = \{x, y, xy, yx\}$ defines the set of gradient directions. The convolution kernels $g_x, g_y, g_{xy}$ and $g_{yx}$ are described as:

$$g_x = \begin{bmatrix} -1 & 1 \end{bmatrix}, \quad g_y = g_x^T,$$
$$g_{xy} = \begin{bmatrix} -1 & 0 \\ 0 & 1 \end{bmatrix}, \quad g_{yx} = \begin{bmatrix} 0 & -1 \\ 1 & 0 \end{bmatrix}. \tag{7}$$

The score measures gradients in horizontal, vertical, and diagonal directions. High scores typically correspond to regions with strong high-frequency variations, which are more likely caused by noise or structural details rather than meaningful image content. In AI-generated images, such high-divergence areas often indicate artifacts resulting from imperfections in the generative models. Therefore, we select the noisy patch with the highest $g_p$ score:

$$\tilde{z}_{p^*} = \arg\max_p g_p, \tag{8}$$

where $p^*$ denotes the index of the best image patch.

Although PatchCraft (Zhong et al., 2023), ESSP (Chen et al., 2024b) and our GBPS all involve patch-based selection, ours distinguishes them in three main aspects. First, GBPS relies on a gradient-based score instead of evaluating texture diversity. Second, our formulation is efficiently implemented through image convolution. Finally, our approach identifies the patch with the maximum score, whereas ESSP opts for the minimum and PatchCraft employs more than a single patch.

**Patch-Based Convolutional Classifier:** After selecting the important image patch, we feed it into a ResNet-50–based convolutional classifier (He et al., 2016), chosen for its simplicity and effectiveness. To adapt ResNet-50 for 32×32 patches, we introduce several modifications to preserve spatial information and prevent premature feature compression. Specifically, we reduce the stride of the initial convolution from 2 to 1, keeping the output resolution at 32×32, and remove the max-pooling layer to avoid downsampling to 16×16. We further modify the first bottleneck block in the second layer by changing the strides of both the 3×3 and the corresponding 1×1 convolutions from 2 to 1, ensuring the output remains 32×32. The third and fourth layers remain unchanged, so the adapted ResNet-50 produces 8×8 features before the final pooling layer. By alleviating aggressive early downsampling, these modifications preserve fine spatial details essential for accurate representation learning while maintaining the hierarchical feature extraction capacity of ResNet-50.

# 4 EXPERIMENTS

## 4.1 DATASETS AND IMPLEMENTATION DETAILS

We conduct extensive experiments on various mainstream datasets, including AIGCDetection-Benchmark (AIGCDB) (Zhong et al., 2023) and GenImage (Zhu et al., 2024). We adopt accuracy (ACC) as the evaluation metric.

**AIGCDB:** AI-generated images in the AIGCDB are generated by 17 GAN-based or Diffusion-based generators. We train the model on the subset ProGAN, and test on all 17 subsets to compute the average accuracy.

**GenImage:** GenImage benchmark is a million-scale dataset especially designed for AI-generated image detection. Real images are sourced from ImageNet dataset (Deng et al., 2009), while AI-generated images are generated by eight mainstream GAN and Diffusion based generators, including BigGAN (Brock et al., 2018), Midjourney (Mid, 2022), Wukong (wuk, 2022), Stable Diffusion V1.4 (Rombach et al., 2022), Stable Diffusion V1.5 (Rombach et al., 2022), ADM (Dhariwal & Nichol, 2021), GLIDE (Nichol et al., 2021), and VQDM (Gu et al., 2022). Following the setting of (Luo et al., 2024; Yan et al., 2025; Chen et al., 2024b), we train the model on subset Stable Diffusion V1.4, and test on all eight subsets.

**Implementation Details:** The image is first resized to 256×256 before construction of bit reversed image. Patches are randomly sampled during training. After the 32×32 patch is selected, it is fed into the subsequent modified ResNet-50 classifier, which is pretrained on ImageNet (Deng et al., 2009). Training is conducted with a maximum of 16 epochs, a batch size of 64, a learning rate of 0.0001, and the Adam optimizer. For zero-shot scenarios, we only utilize real images of ImageNet (Deng et al., 2009). Apart from bit-reversed construction and patch selection, these images are fed into modified ResNet-50 pretrained on bit-reversed images to obtain the average feature after the final global average pooling. Then the testing images are input into the same architecture to obtain testing features for evaluating distances from features of real images.

| Method | AIGCDB | GenImage | | | | | | | | |
|---|---|---|---|---|---|---|---|---|---|---|
| | | BigG | Midj | Wuk | SDV1.4 | SDV1.5 | ADM | GLIDE | VQDM | Avg. |
| Spec (Zhang et al., 2019) | - | 49.8 | 52.0 | 94.8 | 99.4 | 99.2 | 49.7 | 49.8 | 55.6 | 68.8 |
| DeiT-S (Touvron et al., 2021) | - | 53.5 | 55.6 | 98.9 | 99.0 | 99.8 | 49.8 | 58.1 | 56.9 | 71.6 |
| LGrad (Tan et al., 2023) | 75.3 | - | - | - | - | - | - | - | - | - |
| LNP (Liu et al., 2022) | 83.8 | - | - | - | - | - | - | - | - | - |
| CNNSpot (Wang et al., 2020) | 70.8 | 46.8 | 52.8 | 78.6 | 96.3 | 95.9 | 50.1 | 39.8 | 53.4 | 64.2 |
| GramNet (Liu et al., 2020) | 68.4 | 51.7 | 54.2 | 98.9 | 99.2 | 99.1 | 50.3 | 54.6 | 50.8 | 69.9 |
| ResNet-50 (He et al., 2016) | 69.7 | 52.0 | 54.9 | 98.2 | **99.9** | 99.7 | 53.5 | 61.9 | 56.2 | 72.1 |
| GenDet (Liu et al., 2020) | - | 75.0 | 89.6 | 92.8 | 96.1 | 96.1 | 58.0 | 78.4 | 66.5 | 81.6 |
| Swin-T (Liu et al., 2021) | - | 57.6 | 62.1 | 99.1 | **99.9** | 99.8 | 49.8 | 67.6 | 62.3 | 74.8 |
| F3Net (Qian et al., 2020) | - | 49.9 | 50.1 | **99.9** | **99.9** | **99.9** | 49.9 | 50.0 | 49.9 | 68.7 |
| UnivFD (Ojha et al., 2023) | 78.4 | 80.3 | 73.2 | 75.6 | 84.2 | 84.0 | 55.2 | 76.9 | 56.9 | 73.3 |
| PatchCraft (Zhong et al., 2023) | 89.3 | 72.4 | 79.0 | 89.3 | 89.5 | 89.3 | 77.3 | 78.4 | 83.7 | 82.3 |
| DIRE (Wang et al., 2023b) | 67.9 | 72.6 | 58.5 | 58.5 | 99.2 | 95.4 | 61.6 | 79.3 | 49.8 | 72.1 |
| LaRE[2] (Luo et al., 2024) | 54.2 | 63.4 | 84.9 | 83.7 | 99.1 | 99.0 | 90.8 | 92.0 | 64.0 | 84.5 |
| ESSP (Chen et al., 2024b) | 50.1 | 73.9 | 82.6 | 98.6 | 99.2 | 99.1 | 78.9 | 88.9 | 96.0 | 89.7 |
| AIDE (Yan et al., 2025) | 92.8 | 66.9 | 79.4 | 98.7 | 99.7 | 99.8 | 78.6 | 91.8 | 80.3 | 86.9 |
| VIB-Net (Zhang et al., 2025) | - | 95.8 | 61.3 | 75.9 | 71.6 | 70.0 | 71.5 | 69.4 | 86.7 | 84.2 |
| UniFD (Ojha et al., 2023) | 77.1 | 90.0 | 56.1 | 70.7 | 63.6 | 63.9 | 67.6 | 62.7 | 85.6 | 70.1 |
| NPR (Tan et al., 2024b) | 91.7 | 80.7 | 91.7 | 94.0 | 94.4 | 94.4 | 87.8 | 93.2 | 88.7 | 90.6 |
| C2P (Tan et al., 2025) | 96.2 | 98.7 | 88.2 | 98.8 | 90.9 | 97.9 | 96.4 | 99.0 | 96.5 | 95.8 |
| FatFormer (Liu et al., 2024) | 93.3 | 55.8 | 92.7 | 99.9 | 100.0 | 99.9 | 75.9 | 98.9 | 98.8 | 88.9 |
| RAID (ours) | **93.5** | **98.9** | **97.2** | 97.7 | 98.9 | 98.8 | **97.8** | **99.1** | 97.5 | **98.4** |

Table 1: **Evaluation on the datasets of AIGCDB (Zhong et al., 2023) and GenImage (Zhu et al., 2024).** Following existing protocols, for AIGCDB, models are trained on ProGAN and evaluated on all subsets of AIGCDB, with the averaged accuracy reported; for GenImage, models are trained on Stable Diffusion V1.4 and evaluated on all subsets, including BigGAN (BigG), Midjourney (Midj), Wukong (Wuk), Stable Diffusion V1.4 (SDV1.4), Stable Diffusion V1.5 (SDV1.5), ADM, GLIDE, and VQDM.

| Method | BigGAN | Midjourney | Wukong | SDV1.4 | SDV1.5 | ADM | GLIDE | VQDM | Avg. |
|---|---|---|---|---|---|---|---|---|---|
| RIGID (He et al., 2024) | 53.0 | 94.1 | 87.8 | 87.0 | 87.2 | 51.4 | 45.9 | 52.2 | 69.8 |
| AEROBLADE (Ricker et al., 2024) | 58.3 | 40.2 | 51.4 | 52.6 | 55.1 | 50.7 | 29.4 | 52.8 | 48.8 |
| Manifold (Brokman et al., 2025) | 77.6 | 55.5 | 65.4 | 62.0 | 63.0 | 57.3 | 88.3 | 76.9 | 68.2 |
| RAID (ours) | 91.0 | 85.9 | 86.2 | 86.3 | 86.8 | 85.5 | 83.9 | 84.5 | 86.3 |

Table 2: **Zero-shot AI-generated image detection on the GenImage dataset.** We compare our results with other mainstream zero-shot methods on the GenImage.

## 4.2 EVALUATION OF AI-GENERATED IMAGE DETECTION

**Evaluation on the AIGCDB and GenImage:** We compare our results with other mainstream methods on three datasets, and results are shown in Table 1.

On the AIGCDB dataset, our approach surpasses all prevailing methods, achieving an average accuracy of 93%.

On the GenImage dataset, despite marginally inferior performance on a few subsets, our method exhibits extremely strong generalization capabilities on all subsets, and achieves an average accuracy of 98.4%, surpassing SOTA approaches by over 8.7% margin.

**Capability of Generalization:** We compare our approach with four competitive methods (DIRE, ResNet-50, LaRE[2], and ESSP) on eight subsets of GenImage (Zhu et al., 2024). As shown in Figure 5, existing mainstream methods achieve reasonable results only when the training subset and testing subset are identical, and easily encounter difficulties when evaluated on subsets from unseen generators. Despite recent methods (e.g., LaRE[2] and ESSP) mitigating these generalization issues by achieving strong performance on subsets such as Wukong, Stable Diffusion V1.4, and Stable Diffusion V1.5, they still struggle to detect AI-generated images on other subsets, such as BigGAN, ADM, and Midjourney. Remarkably, our method, merely trained on one subset, shows exceptional detecting accuracy on all subsets, exhibiting excellent generalization capability.

**Zero-Shot Generalization Performance:** To further verify the effectiveness of our bit reversed images, we conduct zero-shot AI-generated image detection using only real images from ImageNet for

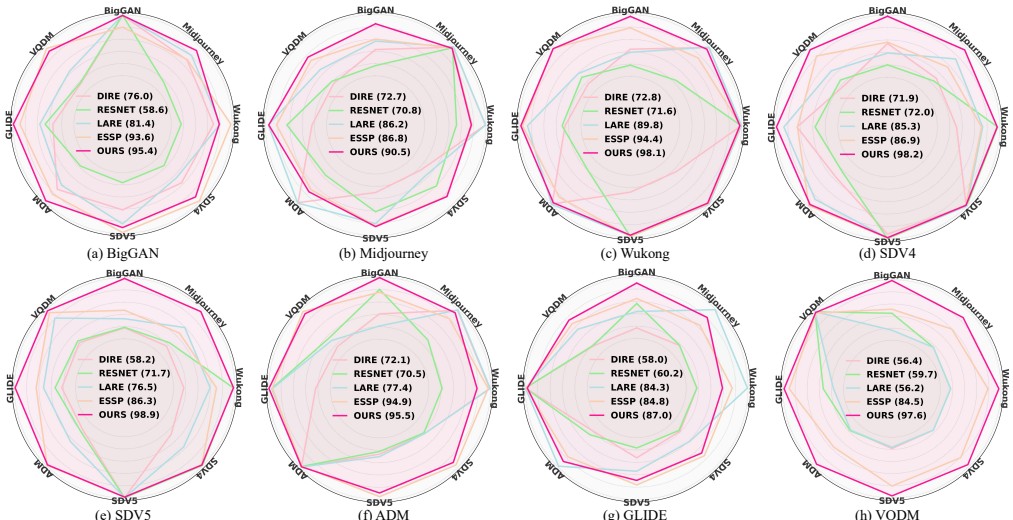

Figure 5: **Evaluation of generalization capability.** Four competitive methods (DIRE, ResNet-50, LaRE[2], and ESSP) and our approach are trained on eight subsets (corresponding to eight subplots) and evaluated on eight subsets (corresponding to eight dimensions of each subplot) of Gen-Image (Zhu et al., 2024).

| Method | BigGAN | Midjourney | Wukong | SDV1.4 | SDV1.5 | ADM | GLIDE | VQDM | Avg. |
|--------|--------|-----------|--------|--------|--------|-----|-------|------|------|
| RAID (ours) | **98.9** | **97.2** | 97.7 | 98.9 | 98.8 | **97.8** | **99.1** | **97.5** | **98.4** |
| w/o BRI | 74.1 | 90.9 | 99.6 | 99.9 | **99.9** | 58.1 | 90.5 | 84.1 | 87.7 |
| w/o GBPS | 57.0 | 58.2 | **99.9** | **100.0** | **99.9** | 55.0 | 58.7 | 60.3 | 74.8 |
| w/o BRI-GBPS | 52.0 | 54.9 | 98.2 | 99.9 | 99.7 | 53.5 | 61.9 | 56.2 | 72.1 |

Table 3: **Ablation studies.** We conduct the ablation studies by removing modules of Bit Reversed Images (BRI), and Gradient-Based Patch Selection (GBPS), respectively.

training. As there is limited prior work on this zero-shot setting, we compare our approach with three methods: RIGID (He et al., 2024), AEROBLADE (Ricker et al., 2024), and Manifold (Brokman et al., 2025). Results on the eight subsets of GenImage (Zhu et al., 2024) are shown in Table 2. Without using AI-generated images for training, our RAID achieves 86.3% average accuracy and strong performance across all eight subsets, whereas other mainstream zero-shot methods perform well on only a few subsets.

### 4.3 ABLATION STUDIES AND ANALYSES

We conduct extensive ablation studies and analyses to validate the effectiveness of our approach. All models are trained on Stable Diffusion V1.4, and evaluated on eight subsets of GenImage: Big-GAN (Big), Midjourney (Mid), Wukong (Wuk), Stable Diffusion V1.4 (SD4), Stable Diffusion V1.5 (SD5), ADM, GLIDE (GLI) and VQDM (VQD).

**Ablation Studies:** We respectively remove modules of Bit Reversed Images (BRI), and Gradient-Based Patch Selection (GBPS) and compare results in Table 3. After removing the BRI and GBPS modules, the average accuracy drops from 98.4% to 87.7% and 74.8%, respectively, highlighting the importance of each module in enhancing detection performance and generalization capability.

**Performances of Different Bit Reversed and Forward Images:** To determine which bit reversed or forward image is the most effective, we compare results of diverse bit reversed and forward images in Table 4. In terms of bit reversed images, ① represents full reversal and exhibits the most competitive results of 98.4%, demonstrating the effectiveness of prioritizing lower-order bit-planes. In terms of bit forward images, average accuracies achieve a high level for variants ④~⑦, when more lower-order bit-planes are moved to higher-order positions. Notably, it reaches the peak of 98.0% for the ⑥.

| Method | | Bit Order: From 0 to 7 | | | | | | | | Big | Mid | Wuk | SD4 | SD5 | ADM | GLI | VQD | Avg. |
|---|---|---|---|---|---|---|---|---|---|---|---|---|---|---|---|---|---|---|
| Bit Reversed Images ① | 7 | 6 | 5 | 4 | 3 | 2 | 1 | 0 | | **98.9** | 97.2 | **97.7** | **98.9** | **98.8** | 97.8 | **99.1** | 97.5 | **98.4** |
| ② | 6 | 5 | 4 | 3 | 2 | 1 | 0 | 7 | | 92.5 | 92.2 | 94.2 | 97.3 | 97.4 | 87.7 | 97.8 | 88.3 | 93.6 |
| ③ | 5 | 4 | 3 | 2 | 1 | 0 | 7 | 6 | | 90.9 | 81.9 | 92.5 | 95.2 | 95.2 | 74.2 | 75.3 | 91.7 | 87.4 |
| ④ | 4 | 3 | 2 | 1 | 0 | 7 | 6 | 5 | | 90.7 | 92.2 | 89.2 | 94.3 | 94.2 | 76.0 | 82.0 | 89.3 | 88.7 |
| ⑤ | 3 | 2 | 1 | 0 | 7 | 6 | 5 | 4 | | 97.7 | **98.9** | 94.5 | 97.9 | 98.2 | 88.2 | 89.7 | 94.3 | 95.1 |
| ⑥ | 2 | 1 | 0 | 7 | 6 | 5 | 4 | 3 | | 97.4 | 97.1 | 95.9 | 97.2 | 97.1 | 97.2 | 97.7 | 96.5 | 97.0 |
| ⑦ | 1 | 0 | 7 | 6 | 5 | 4 | 3 | 2 | | 96.9 | 97.5 | 92.7 | 97.9 | 98.0 | 88.2 | 99.0 | 90.7 | 95.2 |
| ⑧ | 0 | 7 | 6 | 5 | 4 | 3 | 2 | 1 | | 89.8 | 95.5 | 92.4 | 96.4 | 96.3 | 82.0 | 88.3 | 89.0 | 91.4 |
| Bit Forward Images ① | 0 | 1 | 2 | 3 | 4 | 5 | 6 | 7 | | 74.1 | 90.9 | **99.6** | **99.9** | **99.9** | 58.1 | 90.5 | 84.1 | 87.7 |
| ② | 1 | 2 | 3 | 4 | 5 | 6 | 7 | 0 | | 81.4 | 86.5 | 95.1 | 98.3 | 98.6 | 60.1 | 81.7 | 82.4 | 86.0 |
| ③ | 2 | 3 | 4 | 5 | 6 | 7 | 0 | 1 | | 91.1 | 91.9 | 91.9 | 96.6 | 96.6 | 63.5 | 80.6 | 86.2 | 87.7 |
| ④ | 3 | 4 | 5 | 6 | 7 | 0 | 1 | 2 | | 97.3 | 95.2 | 96.5 | 98.8 | 98.7 | 86.6 | 82.3 | 97.1 | 94.3 |
| ⑤ | 4 | 5 | 6 | 7 | 0 | 1 | 2 | 3 | | 96.6 | **98.9** | 94.0 | 98.1 | 98.3 | 83.4 | 83.4 | 93.3 | 93.5 |
| ⑥ | 5 | 6 | 7 | 0 | 1 | 2 | 3 | 4 | | **98.5** | 98.3 | 97.1 | 98.3 | 98.4 | **97.3** | **98.7** | 98.7 | **98.0** |
| ⑦ | 6 | 7 | 0 | 1 | 2 | 3 | 4 | 5 | | 95.2 | 97.7 | 93.6 | 98.0 | 97.8 | 87.2 | **98.7** | 98.8 | 95.0 |
| ⑧ | 7 | 0 | 1 | 2 | 3 | 4 | 5 | 6 | | 90.5 | 93.3 | 98.9 | 99.7 | 99.7 | 77.1 | 98.3 | 82.4 | 92.8 |

Table 4: **Performances of different bit reversed and forward images.** Different bit reversed and forward images are utilized. Our approach is ① of bit reversed images, and ① of bit forward images is the baseline using the original RGB image.

| Method | Gauss-0 | Gauss-1 | Gauss-2 | Gauss-3 | JPEG-100 | JPEG-98 | JPEG-95 | JPEG-90 |
|---|---|---|---|---|---|---|---|---|
| RAID (ours) | 98.4 | 84.8 | 80.4 | 77.5 | 98.4 | 82.3 | 79.5 | 75.6 |
| ESSP (Chen et al., 2024b) | 89.7 | 80.9 | 58.3 | 53.4 | 89.7 | 80.3 | 74.5 | 66.0 |
| UniFD (Ojha et al., 2023) | 70.1 | 64.7 | 64.1 | 61.9 | 70.1 | 65.5 | 64.5 | 62.9 |

Table 5: **Results under image perturbations.** Gaussian blur with different standard deviations and JPEG compression with varying quality levels are applied to the input image.

**Robustness Against Image Degradation:** In Table 5, we show AI-generated image detection results of our approach against two representative types of image degradation: Gaussian blur and JPEG compression. Compared to the single patch-based ESSP (Chen et al., 2024b) and frequency-based UniFD (Ojha et al., 2023), our patch-based method using the bit reversed image is more robust to image degradation and noise perturbation.

**Computation Efficiency:** We compare the computational efficiency of our approach with other mainstream methods (DIRE, LaRE$^2$, and ESSP) in Table 6. For the inference speed, DIRE and LaRE require multiple steps to construct the feature map, resulting in higher latency (1.99 s and 250 ms, respectively), while our approach completes this process in a single step, taking only 2.09 ms. The patch-based method ESSP takes a total of 31.99 ms, while our approach operates at the millisecond level. In terms of model parameters, other mainstream methods mainly rely on large pretrained models (e.g., diffusion), which introduce a substantial number of parameters. In contrast, our approach is significantly more lightweight and efficient, requiring only 23.5 M parameters.

**Visualizations of Fake Probabilities of Patches:** Since our approach is based on single patch of bit reversed image, we visualize the predicted fake probabilities for evenly divided patches in

| Method | Time | | Params | |
|---|---|---|---|---|
| | Feature Extraction | Total | Feature Extraction | Total |
| DIRE (Wang et al., 2023b) | 1.99 s | 2 s | 644.8 M | 688.3 M |
| LaRE$^2$ (Luo et al., 2024) | 250 ms | 260 ms | 1066.2 M | 1165.8 M |
| ESSP (Chen et al., 2024b) | 25.10 ms | 31.99 ms | 7.1 M | 30.7 M |
| RAID (ours) | 2.09 ms | 4.23 ms | 0 | 23.5 M |

Table 6: **Comparison of computation efficiency.** Our approach and other mainstream methods are compared in computation time and model parameters.

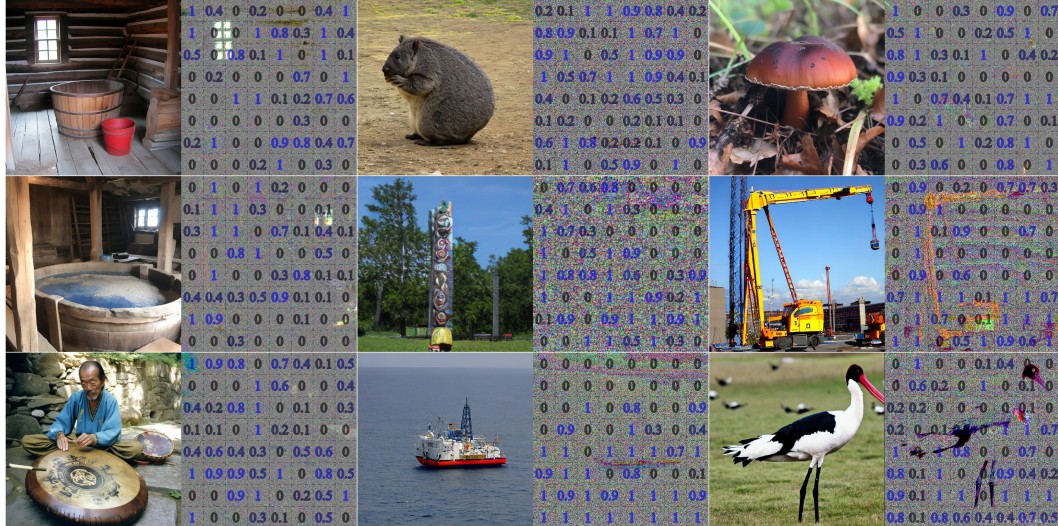

Figure 6: **Visualizations of AI-generated images and the corresponding probabilities of small patches predicted as fake**. The images are resized to 256×256, with a patch size of 32×32, so each image is evenly divided into 8×8 patches. For each sample, the original RGB image is on the left, and fake probabilities on the corresponding bit reverse image are on the right. The probability is at the center of each patch, with more intense blue colors representing higher probabilities. Regions with probabilities greater than 0.5 indicate patches that are successfully predicted as fake.

Figure 6. We select AI-generated images that appear highly realistic, and visualize the predicted fake probabilities of patches of bit revised images. We find that artifacts in AI-generated images are invisible in the original images, but become visible in bit reversed images and can be successfully detected by our model.

## 5 CONCLUSION

In this paper, we studies AI-generated image detection from the perspective of bit-planes and introduce an innovative representation named bit reversed image. The bit reversed image is a reversible encoding of the original RGB image, but it evidently amplifies artifacts that are invisible in the original image. Following this insight, we propose a simple yet highly effective approach for AI-generated image detection. Extensive experiments, including cross-generator evaluation, cross-dataset evaluation, and zero-shot AI-generated image detection, consistently demonstrate the effectiveness of our approach. In addition, it contains only 23.5 million parameters and runs in milliseconds. One limitation of our approach is that the deepfake image classification model we use is the standard ResNet. We will design a more tailored architecture by incorporating the characteristics of bit reversed images.

**Limitation Discussion:** Although our method can reveal artifacts hidden in the low-bit planes of visually realistic AI-generated images and detect them accordingly, it relies solely on low-level noise information rather than high-level semantic cues. In addition, as low-bit planes are easily affected by environmental factors, such as dim lighting or uniform backgrounds, our model may fail under such conditions.

## ETHICS STATEMENT

This work proposes a method for detecting AI-generated images, with potential positive impact in enhancing deepfake detection and preventing misuse of generative models. A possible risk is that the method could be exploited to improve deepfake generation. To mitigate this, we plan to adopt responsible release practices.

## REPRODUCIBILITY STATEMENT

All implementation details, datasets, and hyperparameters are described in the paper and supplementary material. We will release the code and pretrained models to ensure full reproducibility of our results.

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

APPENDIX

# A    NEW DATASETS FOR AI-GENERATED IMAGE DETECTION

## A.1    DATASET DESCRIPTION

**GID:** We construct a new dataset to further evaluate the performance. Real images are sourced from ImageNet (Deng et al., 2009), and the AI-generated images are generated by currently competitive generators: Google Imagen 2, FLUX.1, DALL-E 3, Stable Diffusion 3, and WANX 2.1.

**GVD:** We also construct a challenging benchmark for AI-generated image detection, employing AI-generated images extracted from AI-generated videos, which are generated by state-of-the-art video generative models including MuseV (Xia et al., 2024), SVD (Blattmann et al., 2023), Mora (Yuan et al., 2024), CogVideo (Hong et al., 2022), Text2Video-Zero (Khachatryan et al., 2023), Tune-A-Video (Wu et al., 2023), and VideoCrafter2 (Chen et al., 2024a). Real images are extracted from videos including HD-VG (Wang et al., 2023a), Youtube, and Bilibili.

**Data Distribution of GID and GVD:** We primarily use hard cases of AI-generated detection (Ni et al., 2025), such as plants, vehicles, people, buildings, natures, etc, with similar sample distribution across these categories. These datasets cover challenging scenarios such as low-illumination scenes (16.6%, including twilight, dawn, nighttime, low-light environments, etc.), fast-moving objects (22.4%, including vehicles, human motions, animal movements, natural phenomena, etc), and extreme environment scenes (4.9%, including high-risk challenges, aerial activities, etc.). Details about our proposed datasets are summarized in Table 7.

| Dataset | Subset | | Generator | Label | Images |
|---|---|---|---|---|---|
| | Imagen 2 | Imagen 2 | Google Imagen 2 | Fake | 2,000 |
| | | Real | — | Real | 2,000 |
| | FLUX.1 | FLUX.1 | FLUX.1 | Fake | 2,000 |
| | | Real | — | Real | 2,000 |
| GID | DALL-E 3 | DALL-E 3 | DALL-E 3 | Fake | 2,000 |
| | | Real | — | Real | 2,000 |
| | SD3 | SD3 | Stable Diffusion 3 | Fake | 2,000 |
| | | Real | — | Real | 2,000 |
| | WANX 2.1 | WANX 2.1 | WANX 2.1 | Fake | 2,000 |
| | | Real | — | Real | 2,000 |
| | Group 1 | MuseV | MuseV | Fake | 10,000 |
| | | SVD | Diffusion | Fake | 10,000 |
| | | CogV | CogVideo | Fake | 10,000 |
| | | Mora | Mora | Fake | 10,000 |
| GVD | | HD-VG | — | Real | 40,000 |
| | Group 2 | COG | CogVideo | Fake | 2,500 |
| | | T2VZ | Text2Video-Zero | Fake | 2,500 |
| | | TAV | Tune-A-Video | Fake | 2,500 |
| | | VC | VideoCrafter | Fake | 2,500 |
| | | YT-BI | — | Real | 10,000 |

Table 7: **A summary of the introduced datasets: GID and GVD.** GID comprises five subsets: Google Imagen 2 (Imagen 2), FLUX.1, DALL-E 3, Stable Diffusion 3 (SD3), and WANX 2.1. GVD comprises two groups, and each group contains four fake subsets and one real subset.

## A.2    CROSS-DATASET EVALUATION

**Evaluation on the GID:** We train the model on Stable Diffusion V1.4 from GenImage, and evaluate it on Google Imagen 2 (Image 2), FLUX.1, DALL-E 3, Stable Diffusion 3 (SD3), and WANX 2.1. On the proposed GID dataset, we reproduce the results of recent AI-generated image detection

methods, including LGrad (Tan et al., 2023), LNP (Liu et al., 2022), CNNSpot (Wang et al., 2020), GramNet (Liu et al., 2020), UnivFD (Ojha et al., 2023), DIRE (Wang et al., 2023b), LaRE$^2$ (Luo et al., 2024), AIDE (Yan et al., 2025), and ESSP (Chen et al., 2024b). As shown in Table 8, most methods face substantial difficulties, with their accuracy dropping to around 50%, which is equivalent to random guessing, highlighting the formidable challenge posed by the GID dataset. Although recent approaches AIDE and ESSP exceed 80% accuracy, they still significantly underperform our method achieving 97.7%.

| Method | Imagen 2 | FLUX.1 | DALL-E 3 | SD3 | WANX 2.1 | Avg. |
|---|---|---|---|---|---|---|
| LGrad (Tan et al., 2023) | 75.7 | 57.0 | 77.2 | 82.5 | 76.1 | 73.7 |
| LNP (Liu et al., 2022) | 82.4 | 54.9 | 9.1 | 52.1 | 43.7 | 48.4 |
| CNNSpot (Wang et al., 2020) | 50.5 | 49.2 | 59.1 | 66.2 | 51.0 | 55.2 |
| GramNet (Liu et al., 2020) | 64.4 | 49.4 | 56.8 | 70.3 | 64.4 | 61.1 |
| ResNet-50 (He et al., 2016) | 54.9 | 51.8 | 63.6 | 67.5 | 61.6 | 60.0 |
| UnivFD (Ojha et al., 2023) | 49.7 | 69.2 | 50.0 | 58.7 | 50.8 | 55.7 |
| DIRE (Wang et al., 2023b) | 51.4 | 50.0 | 51.0 | 77.2 | 46.9 | 55.3 |
| LaRE$^2$ (Luo et al., 2024) | 57.9 | 81.1 | 59.1 | 60.0 | 50.5 | 61.7 |
| ESSP (Chen et al., 2024b) | 96.7 | 84.3 | 81.8 | 95.2 | 83.8 | 88.4 |
| AIDE (Yan et al., 2025) | 87.3 | 91.0 | 95.1 | 86.6 | 70.3 | 86.1 |
| RAID (ours) | **98.8** | **99.4** | **97.4** | **98.9** | **94.2** | **97.7** |

Table 8: **Cross-dataset evaluation on the proposed GID dataset.** Models are trained on the SDV1.4 subset of the GenImage (Zhu et al., 2024), and directly evaluated on Google Imagen 2 (Imagen 2), FLUX.1, DALL-E 3, Stable Diffusion 3 (SD3), and WANX 2.1. We reproduce the results of different compared methods.

**Evaluation on the GVD:** We train the model on an image-based dataset, specifically using the Stable Diffusion V1.4 subset from GenImage, and test it on the GVD dataset, which is collected from challenging videos. We evaluate LaRE$^2$, AIDE, ESSP and our method, all achieving high detecting accuracy on GenImage, on our challenging benchmark GVD. As shown in Table 9, all methods encounter a significant drop on GVD. That demonstrate existing methods have weak capacity for detecting AI-generated frames extracted from AI-generated videos, which substantially validates the formidable challenge posed by our proposed dataset GVD for AI-generated image detection. Nonetheless, our approach evidently surpasses prevailing methods.

| Group 1 | MuseV | SVD | Mora | CogV | HD-VG | Avg. |
|---|---|---|---|---|---|---|
| LaRE$^2$ (Luo et al., 2024) | 7.1 | 6.8 | 23.6 | 37.5 | 63.8 | 41.3 |
| ESSP (Chen et al., 2024b) | 33.4 | 38.4 | 32.6 | 39.8 | 61.5 | 48.8 |
| AIDE (Yan et al., 2025) | 15.7 | 17.3 | 27.2 | 23.2 | 75.2 | 48.0 |
| RAID (ours) | **44.5** | **53.6** | **62.4** | **62.4** | **77.2** | **66.5** |

| Group 2 | COG | T2VZ | TAV | VC | YT-BI | Avg. |
|---|---|---|---|---|---|---|
| LaRE$^2$ (Luo et al., 2024) | 13.1 | 15.8 | 32.1 | **38.0** | 55.8 | 40.3 |
| ESSP (Chen et al., 2024b) | 28.8 | 18.8 | 15.4 | 19.7 | 47.4 | 34.0 |
| AIDE (Yan et al., 2025) | 2.4 | **40.2** | 37.4 | 23.0 | 76.3 | 51.0 |
| RAID (ours) | **35.1** | 20.2 | **64.6** | 34.8 | **79.5** | **59.1** |

Table 9: **Cross-dataset evaluation on the GVD dataset.** We compare our results with other mainstream methods on GVD.

# B  ADDITIONAL EXPERIMENTS AND VISUALIZATIONS

## B.1  IMPACT OF DIFFERENT WEIGHTS

Table 10 further investigates the impacts of different weights while constructing bit reversed images in Eq. (1). Generally, we assign most significant weights to the lowest-order bit-planes (such as

| | 0 | 1 | 2 | 3 | 4 | 5 | 6 | 7 | Big | Mid | Wuk | SD4 | SD5 | ADM | GLI | VQD | Avg. |
|---|---|---|---|---|---|---|---|---|---|---|---|---|---|---|---|---|---|
| ① | 128 | 64 | 32 | 16 | 8 | 4 | 2 | 1 | 98.9 | 97.2 | 97.7 | 98.9 | 98.8 | 97.8 | 99.1 | 97.5 | 98.4 |
| ② | 48 | 48 | 48 | 48 | 32 | 16 | 8 | 4 | 90.4 | 93.5 | 91.5 | 94.4 | 93.4 | 93.0 | 94.6 | 97.7 | 91.3 |
| ③ | 64 | 64 | 64 | 32 | 16 | 8 | 4 | 2 | 94.1 | 95.9 | 94.9 | 96.3 | 96.2 | 87.5 | 95.8 | 92.2 | 94.2 |
| ④ | 32 | 64 | 64 | 64 | 16 | 8 | 4 | 2 | 92.9 | 96.9 | 93.0 | 95.3 | 95.4 | 85.2 | 96.5 | 90.6 | 93.3 |
| ⑤ | 96 | 96 | 32 | 16 | 8 | 4 | 2 | 1 | 99.4 | 98.5 | 99.0 | 99.6 | 99.5 | 99.5 | 99.6 | 99.0 | 99.3 |
| ⑥ | 32 | 96 | 96 | 16 | 8 | 4 | 2 | 1 | 98.8 | 98.3 | 98.2 | 98.9 | 99.0 | 98.5 | 99.1 | 98.0 | 98.6 |
| ⑦ | 32 | 16 | 96 | 96 | 8 | 4 | 2 | 1 | 94.6 | 97.6 | 93.5 | 96.5 | 96.6 | 86.0 | 96.3 | 91.7 | 94.2 |
| ⑧ | | | | Learnable | | | | | 85.6 | 70.1 | 94.7 | 95.4 | 95.1 | 66.5 | 91.2 | 57.0 | 82.2 |

Table 10: **Impact of different weights of bit-planes.** Different and learnable weights are utilized during construction of bit reversed images. The column index denotes the position of bit-planes, and circled indices are different variants of our approach.

| Trainning | Big | Mid | Wuk | SD4 | SD5 | ADM | GLI | VQD | Avg. |
|---|---|---|---|---|---|---|---|---|---|
| Big | 99.6 | 94.9 | 86.0 | 92.5 | 92.7 | 99.3 | 99.7 | 93.4 | 94.7 |
| Mid | 84.5 | 96.3 | 79.2 | 84.1 | 84.9 | 79.2 | 93.2 | 99.3 | 85.0 |
| Wuk | 98.8 | 95.3 | 98.1 | 98.4 | 98.4 | 97.7 | 98.4 | 98.1 | 97.9 |
| SD4 | 98.4 | 96.8 | 97.5 | 98.4 | 98.4 | 97.7 | 98.6 | 97.5 | 98.0 |
| SD5 | 97.8 | 95.9 | 96.1 | 97.6 | 97.8 | 96.9 | 98.4 | 96.2 | 97.1 |
| ADM | 99.7 | 95.6 | 88.1 | 93.8 | 93.8 | 99.8 | 99.8 | 94.7 | 95.6 |
| GLI | 83.9 | 82.8 | 66.6 | 71.4 | 71.2 | 81.9 | 98.7 | 95.2 | 78.7 |
| VQD | 99.3 | 94.8 | 98.9 | 99.1 | 99.0 | 98.4 | 99.1 | 99.6 | 98.5 |
| Average | 95.3 | 94.1 | 88.8 | 91.9 | 92.0 | 93.9 | 98.2 | 96.8 | 93.2 |

Table 11: **Cross-generator performance on the GenImage dataset.** We train our model on eight subsets of GenImage subsets respectively, and each model is evaluated on these eight subsets. The detection accuracy of both training subsets and testing subsets is averaged. For conciseness, Big-GAN, Midjourney, Wukong, Stable Diffusion V1.4, Stable Diffusion V1.5, ADM, GLIDE, and VQDM are denoted as Big, Mid, Wuk, SD4, SD5, ADM, GLI, and VQD, respectively.

0∼1, 0∼2, 1∼3, etc) based on results of Table 4. The variant ① denotes our approach, while ⑧ is the variant with learnable weights during training. Comparing ① with ②∼④, we observe that increasing the weights of higher bit-planes leads to a noticeable degradation in performance. In contrast, while comparing ① with ⑤, we find that increasing the weights of lower bit-planes slightly increases the performance. These experiments further highlight the critical role of low-bit planes for AI-generated image detection. Although the variant of learnable weights eliminates the need for handcrafted weights and manual tuning, it performs worse than the others, possibly due to overfitting and the model not recognizing the importance of lower bit-planes. Although ⑤ performs even better than ours (①), we use the default weights in our approach for simplicity.

## B.2 CROSS-GENERATOR PERFORMANCE ON THE GENIMAGE

We train our model on eight subsets of GenImage (Zhu et al., 2024), and evaluate each model on eight subsets, respectively. As shown in Table 11, all of our models achieve an average accuracy of exceeding 90%, except for the model trained on GLIDE attains the averaged performance of 87.0%, possibly due to different feature distribution after bit-reversal. Notably, models encounter a slight decline when evaluated on the Wukong, demonstrating the challenge inherent in this subset.

## B.3 VISUALIZATIONS OF FAILURE CASES

Figure 7 visualizes failure cases of our AI-generated image detection approach. The images in the first two rows are real images from ImageNet, but they are incorrectly classified as fake. The images in the last two rows are AI-generated, yet they are wrongly regarded as real. We observe that these hard examples often have dark or poorly lit backgrounds, uniform background colors, or low image quality. These factors affect the noise distribution in the low bit-planes, leading our model to make wrong predictions.

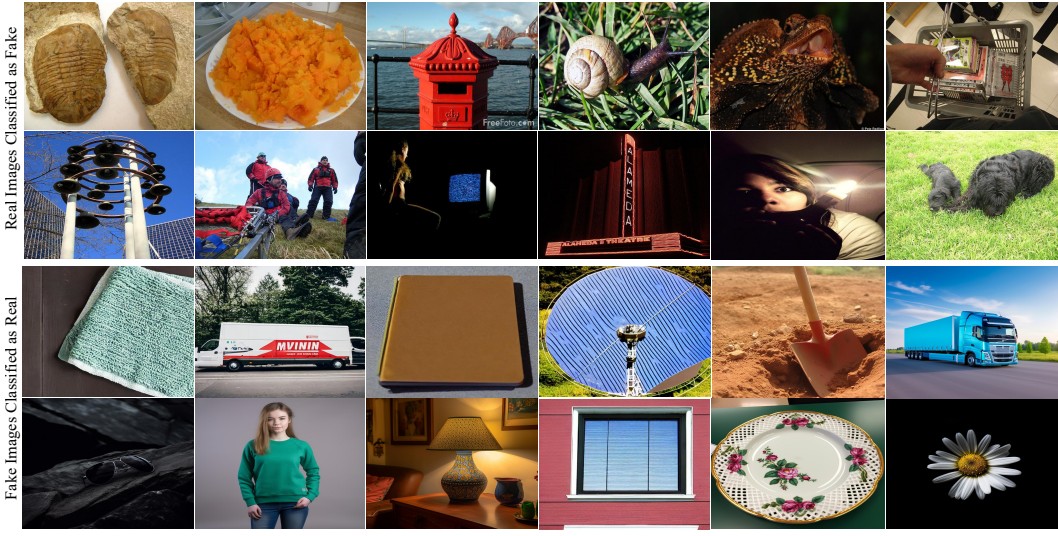

Figure 7: **Visualizations of failure cases of AI-generated image detection**. We visualize both false positive and false negative predictions of our AI-generated image detection method.

