# OpenReview forum: "RAID: Towards Robust AI-Generated Image Detection with Bit Reversed Images"
_ICLR.cc/2026/Conference — Submitted to ICLR 2026_

### Official Review · Reviewer_iJPB · 2025-11-01

**Soundness:** 3
**Presentation:** 3
**Contribution:** 4
**Rating:** 6
**Confidence:** 5

**Summary:**

This paper proposes RAID, a novel framework for AI-generated image detection leveraging bit-plane decomposition and the introduction of Bit Reversed Images (BRI). By reversing bit-plane order, the method amplifies subtle artifacts invisible in original images. Combined with a gradient-based patch selection and a lightweight modified ResNet-50, RAID achieves strong cross-generator and zero-shot generalization across 40+ benchmarks.

**Strengths:**

- The idea of using bit-plane reversal for AI-generated image detection is fresh and unexplored. It introduces a simple yet conceptually interesting perspective beyond spatial and frequency domains. The method reinterprets bit-plane representation — commonly used in steganography — into a discriminative cue for forgery detection, showing creative cross-domain thinking.
Moreover, the integration with a gradient-based patch selector (GBPS) is efficient and complements the novel bit-reversal transformation.
- The experiments are extensive, covering multiple datasets (AIGCDB, GenImage, GID, GVD) and diverse setups (cross-generator, zero-shot, robustness).
- Quantitative results are strong, with clear ablations demonstrating the role of each component.
- The paper is generally well-structured, with detailed methodology and visualization of bit-planes, bit-reversed effects, and patch-level predictions.
- Figures (e.g., Fig. 1–5) are informative and aid understanding of how bit reversal exposes artifacts.
- Given the current importance of detecting synthetic images, this work contributes a lightweight, interpretable, and effective detector. Its generalization to unseen generators and real-world degradation scenarios underscores robustness and potential for deployment.

**Weaknesses:**

- The paper lacks a clear theoretical justification for why bit reversal amplifies AI-specific artifacts. There is no spectral or statistical evidence to show that reversing bit-plane order meaningfully highlights generative inconsistencies rather than generic noise.
- Some strong recent baselines (e.g., C2P-CLIP, NPR, FatFormer, CoD) are missing, limiting the claim of state-of-the-art performance.
- The comparison under image perturbations (Table 6) includes only ESSP, while stronger pretrained universal detectors (e.g., UnivFD) are not discussed.
- Table 4 shows certain bit-forward configurations nearly match bit-reversed performance (98.0% vs. 98.4%), raising questions about the true necessity of reversal. The paper should clarify why full reversal is superior beyond empirical coincidence.
- It remains unclear whether bit reversal corresponds to frequency inversion (i.e., swapping high- and low-frequency components). Without clarifying this, it is hard to position the method relative to existing frequency-based approaches.

**Questions:**

- Can the authors theoretically or empirically justify why bit reversal highlights generative artifacts? Is there a measurable spectral or statistical property supporting this observation?
- Does the bit reversal operation correspond to swapping high- and low-frequency components in the frequency domain? If not, how does it differ from such transformations?
- Table 4 shows bit-forward images can also reach 98.0% accuracy. Why, then, is full reversal necessary?
- Why are comparisons under image perturbations limited to ESSP? How does RAID perform against pretrained universal detectors such as UnivFD, CoD, or FatFormer?
- Could combining BRI with multi-scale or transformer-based architectures further enhance robustness or interpretability?
- What I care most about is what traces the bit-reversed image extracts and its relationship with frequency inversion (i.e., swapping high- and low-frequency components). I will adjust the score based on the author's response.

---

> ### Author Response · Authors · 2025-12-04
> **Response to W1 & Q1**
>
> >**W1: The paper lacks a clear theoretical justification for why bit reversal amplifies AI-specific artifacts. There is no spectral or statistical evidence to show that reversing bit-plane order meaningfully highlights generative inconsistencies rather than generic noise.**
>
> > **Q1: Can the authors theoretically or empirically justify why bit reversal highlights generative artifacts? Is there a measurable spectral or statistical property supporting this observation?**
>
> We thank the reviewer for this insightful question. **We update the paper and add Figure 2, which presents measurable statistical evidence supporting this observation by visualizing the feature distributions of different bit-plane images across real and generative images.** By examining these distributions, we clearly observe that AI-generated images and real images differ markedly at the LSB level: feature points of real images are highly clustered, while those of AI-generated images exhibit noticeably different distributions. In contrast, higher bit-planes show no significant difference.
> This observation supports the necessity of bit reversal in highlighting generative artifacts.
>
> The evidence that the low-bit planes of AI-generated and real images differ can be interpreted as follows:
> First, AI-generated images lack true physical sensor noise. The LSB bit-planes of real images look random, while AI-generated images show structured or unnatural randomness.
> Second, AI-generated images exhibit unnatural high-frequency statistics due to model smoothing. Real images contain rich high-frequency components caused by natural lighting variations, sensor noise and microtextures. Generative models often apply techniques like denoising, upsampling and smoothing inductive biases.
> Third, semantic structures leak into low bit-planes. Neural networks learn semantic correlations during image synthesis. Thus, even the least significant bits may contain faint textures aligned with edges or patterned correlations across pixels. However, LSBs of real images have zero semantic meaning; they are pure noise.
> As a result, low bit-planes from AI images show structure, whereas real images show independent random noise. From the visualizations in Figure 1 and Figure 6, we can observe the above differences in the bit-reversed images. For example, in the rightmost image of the second row in Figure 6, we can clearly observe that semantic structures leak into the low bit-planes of AI-generated images.
>
> Thus, reversing bit weights systematically amplifies the subtle differences in low-order bit-planes between real and AI-generated images while suppressing high-order planes that carry generic structural content.

---

> ### Author Response · Authors · 2025-12-04
> **Response to W5 & Q2 & Q6**
>
> >**W5: It remains unclear whether bit reversal corresponds to frequency inversion (i.e., swapping high- and low-frequency components). Without clarifying this, it is hard to position the method relative to existing frequency-based approaches.**
>
> >**Q2: Does the bit reversal operation correspond to swapping high- and low-frequency components in the frequency domain? If not, how does it differ from such transformations?**
>
> >**Q6: What I care most about is what traces the bit-reversed image extracts and its relationship with frequency inversion (i.e., swapping high- and low-frequency components).**
>
> Bit reversal is fundamentally different from swapping high- and low-frequency components. Frequency-domain transforms (e.g., FFT) decompose an image into a basis of sinusoidal waves, where "high-frequency" refers to rapid spatial oscillations. Swapping them is a global operation that structurally alters the image's frequency composition.
>
> In contrast, our bit-plane decomposition is a numerical​ operation. It slices the image based on the significance of bits in its binary pixel representation. High-order bit-planes carry the major structural information (analogous to low-frequency energy), while low-order bit-planes encode fine details and noise (analogous to high-frequency components, but emphasizing stochasticity).
>
> **What BRI Extracts: Amplified Artifacts:** The Bit-Reversed Image (BRI) does not invert frequencies. Instead, it reallocates weights​ to prioritize lower-order bit-planes. This operation suppresses the "deceptively realistic" structural information in high-order planes and amplifies the subtle, often unnatural, noise patterns and artifacts in the low-order planes that are characteristic of AI generators. These artifacts become strikingly visible in the BRI.
>
> **Key Advantage - Preserved Spatial Locality:** This is a critical advantage over frequency-based methods. Techniques like FreqNet rely on global spectral analysis, which can be sensitive to perturbations like JPEG compression that alter the frequency spectrum. Our BRI, however, operates directly on the pixel domain. By preserving spatial locality, it robustly captures local artifact patterns specific to generators, making the detector less vulnerable to global image manipulations.

---

> ### Author Response · Authors · 2025-12-04
> **Response to Q3 & W4**
>
> >**Q3: Table 4 shows bit-forward images can also reach 98.0% accuracy. Why, then, is full reversal necessary?**
>
> >**W4: Table 4 shows certain bit-forward configurations nearly match bit-reversed performance (98.0% vs. 98.4%), raising questions about the true necessity of reversal. The paper should clarify why full reversal is superior beyond empirical coincidence.**
>
> While certain bit-forward configurations (like variant ⑥ in Table 4) can achieve high accuracy, the full bit-reversal​ (variant ①) is chosen for its superior and more consistent generalization​ and robustness, which are critical for real-world application.
>
> The primary reason is that **full reversal most effectively suppresses the high-order bit-planes that carry the "deceptively realistic" structural content**. This forces the model to rely exclusively on the most subtle artifacts found in the low-order planes, which are less likely to be consistently mimicked by diverse generators. While a specific bit-forward shift might perform well on a given dataset, its performance can be more volatile across different generators, as seen in the varying results for variants ② through ⑦. In contrast, full reversal provides a more principled and generator-agnostic transformation.
>
> **This leads to stronger generalization.** As shown in Figure 4, our method (using full reversal) demonstrates exceptionally balanced and high performance across all eight generators in the GenImage benchmark. A bit-forward image might overfit to artifacts from a specific family of models seen during training. Furthermore, by systematically amplifying the noise in the lowest bit-planes, the full reversal operation enhances robustness against image perturbations like blurring or compression, as these operations directly alter the fine-grained details we highlight.
>
> In summary, full reversal is not strictly necessary for peak performance on a single benchmark, but it is the most reliable and robust strategy for creating a universal detector that performs consistently well in diverse and challenging scenarios.

---

> ### Author Response · Authors · 2025-12-04
> **Response to Q4 & W2**
>
> >**W2: Some strong recent baselines (e.g., C2P-CLIP, NPR, FatFormer, CoD) are missing, limiting the claim of state-of-the-art performance.**
>
> >**Q4: Why are comparisons under image perturbations limited to ESSP? How does RAID perform against pretrained universal detectors such as UnivFD, CoD, or FatFormer?**
>
> We thank the reviewer for the helpful suggestion. Following your advice, **we add comparisons with several strong recent baselines, including C2P-CLIP, NPR, FatFormer, and CoD.**
> We update Table 1 in the manuscript, and the comparison results with these baselines are shown below.
> **The results demonstrate the superior performance of RAID on both AIGCDB and GenImage.**
>  Specifically, RAID achieves the highest average accuracy of 98.4%​ on the GenImage subsets, significantly outperforming UnivFD (70.1%), NPR (90.6%), C2P (95.8%), and FatFormer (88.9%). This notable gap underscores RAID's enhanced generalization capability against diverse generative models.
>
> Delving into individual subsets, RAID maintains robust accuracy above 97% on most tasks, such as BigGAN (98.9%), Midjourney (97.2%), and GLIDE (99.1%). In contrast, other methods exhibit considerable volatility; for instance, FatFormer drops to 55.8% on BigGAN despite high scores on SDV4 and SDV5, indicating overfitting to specific generators. Similarly, UnivFD struggles with Midjourney (56.1%) and ADM (67.6%), highlighting its limited adaptability. Even C2P, while competitive, falls short of RAID's consistency, particularly on ADM (96.4% vs. RAID's 97.8%).
>
>
> | Method   | AIGCDB | BigGAN | Mid  | Wuk | SDV4  | SDV5  | ADM  | GLIDE | VQDM | Avg |
> |    :----:  |  :----:  |  :----:  |  :----: | :----: |  :----:  |  :----:  |  :----:  | :----:  |:----:  |:----:  |
> |   UniFD    |  77.1  |  90.9  |  56.1  |  70.7  |  63.6  |  63.9  |  67.6  |  62.7  |  85.6  |  70.1  |
> |   NPR    |  91.7  |  80.7  |  91.7  |  94.0  |  94.4  |  94.4  |  87.8  |  93.2  |  88.7  |  90.6  |
> |   C2P    |  96.2  |  98.7  |  88.2  |  98.8  |  90.9  |  97.9  |  96.4  |  99.0  |  96.5  |  95.8  |
> |   FatFormer    |  93.3  |  55.8  |  92.7  |  99.9  |  100.0  |  99.9  |  75.9  |  98.9  |  98.8  |  88.9  |
> |   RAID (ours)    |  93.5  |  98.9  |  97.2  |  97.7  |  98.9  |  98.8  |  97.8  |  99.1  |  97.5  |  98.4  |

---

> ### Author Response · Authors · 2025-12-04
> **Response to Q5**
>
> >**Q5: Could combining BRI with multi-scale or transformer-based architectures further enhance robustness or interpretability?**
>
> Thank you for this insightful question. Indeed, exploring the integration of our Bit Reversed Images (BRI) with multi-scale or transformer-based architectures is a promising direction that aligns with our future work considerations mentioned in the conclusion.
>
> Firstly, mutiscale could improve the robustness to images with different resolutions. We have produced this experiment in the similar case, where the low-bit patches are applied. Typically, the accuracy of AI-generated image detection improves with higher input image resolution. At a fixed resolution, accuracy initially increases as the image patch size grows, but begins to decline when the patch size becomes excessively large. The image patch size yielding the highest accuracy is approximately one-eighth of the image resolution. While higher resolutions may improve accuracy, they come at the cost of significantly increased computational overhead. To strike a balance between performance and computational efficiency, we employ an image resolution of 256x256 and an image patch size of 32x32.
>
> | Resolution | 16×16 | 32×32 | 48×48 | 64×64 |
> |:------:|:-------:|:-------:|:-------:|:-------:|
> | 128×128   | **87.3** | 78.4 | 19.8 | 10.4 |
> | 256×256   | 22.4 | **96.8** | 33.1 | 12.3 |
> | 512×512   | 0.19 | 53.5 | **98.9** | 89.7 |
> | 1024×1024   | 75.4 | 93.0 | 98.9 | **100.0** |
>
> Secondly, Transformer can be applied in our method and achieve excellent performance. We crop the bit-reversed image into several patches, which are then selected by Transformer, followed by a simple classifier. The average accuracy on GenImage achieves 98.4\%, demonstrating high compatibility and applicability of RAID.

---

> ### Author Response · Authors · 2025-12-04
> **Response to W3**
>
> >**W3: The comparison under image perturbations (Table 6) includes only ESSP, while stronger pretrained universal detectors (e.g., UnivFD) are not discussed.**
>
> In Table 6 of the updated submission, we compare ESSP, UniFD, and our proposed RAID under various image perturbations. While other methods exhibit notable degradation as perturbations intensify, our RAID consistently maintains stronger robustness. For example, under the strongest Gaussian noise (Gauss-3), RAID achieves 77.5% accuracy, significantly outperforming ESSP (53.4%) and UniFD (61.9%). Similarly, with heavy JPEG compression (JPEG-90), RAID attains 75.6%, well above ESSP (66.0%) and UniFD (62.9%). These results clearly demonstrate RAID’s superior stability and generalization capability, even as image quality degrades.
>
> | Method | Gauss-0 | Gauss-1 | Gauss-2 | Gauss-3 | JPEG-100 | JPEG-98 | JPEG-95 | JPEG-90 |
> |:------:|:-------:|:-------:|:-------:|:-------:|:--------:|:--------:|:--------:|:--------:|
> | RAID (ours)   | 98.4    | 84.8    | 80.4    | 77.5    | 98.4     | 82.3     | 79.5     | 75.6     |
> | ESSP   | 89.7    | 80.9    | 58.3    | 53.4    | 89.7     | 80.3     | 74.5     | 66.0     |
> | UniFD  | 70.1    | 64.7    | 64.1    | 61.9    | 70.1     | 65.5     | 64.5     | 62.9     |

---

### Official Review · Reviewer_JYbp · 2025-11-01

**Soundness:** 3
**Presentation:** 3
**Contribution:** 3
**Rating:** 4
**Confidence:** 4

**Summary:**

This paper proposes RAID, a robust AI-generated image detector that leverages bit-plane decomposition, specifically introducing “bit reversed images” as feature amplifiers for distinguishing real from synthetic imagery. The approach involves lossless bit-plane decomposition, strategic reordering to amplify high-frequency details, gradient-based patch selection for focusing on artifact-rich regions, and a modified ResNet-50 classifier. Extensive experiments are conducted on over 40 benchmarks (AIGCDB, GenImage, and new datasets), with comprehensive ablations and cross-generator/cross-dataset/zero-shot generalization analyses. RAID demonstrates superior performance and is computationally efficient.

**Strengths:**

1. The paper is among the first to successfully operationalize bit-plane analysis and bit reversal as a signal for robust artifact amplification, providing a refreshingly simple yet powerful perspective in the deepfake detection landscape.
2. The construction of bit reversed images, followed by gradient-based patch selection and adapted ResNet-50 deployment, is both theoretically sound and easy to implement. The method’s fast, non-parametric upfront steps (bit-reversal and patch selection) are especially appealing for scalability.

**Weaknesses:**

1. The paper omits a discussion and experimental comparison with “LOTA: Bit-Planes Guided AI-Generated Image Detection” , which is highly relevant, as it also explores bit-plane signals for AI-image detection. The absence of this reference undermines the claim of being among the first to leverage bit-planes and makes it harder to gauge the true incremental value over related work. This missing context should be addressed both in the Related Work section and, ideally, with an experimental baseline.
2. While the intuition for artifact amplification via bit reversal is plausible and empirical results are strong, Section 3.1 could benefit from a deeper mathematical analysis or visualization of why and how bit reversal leads to more pronounced differences in the artifacts between real and AI-generated images. For instance, it would be useful to connect this operation more formally to the types of noise and structure generated by different models.
3. Although RAID is benchmarked against several prior patch-based and frequency/spatial methods, comparison to newer or alternative handcrafted artifact detectors—such as those utilizing global color analysis, learned mask regularization, or ensemble patch aggregation—appears limited. This is especially pertinent for Table 1 and Table 2, which emphasize detection efficacy and generalization, and could be expanded with additional strong baselines.
4. For zero-shot and cross-dataset generalization, relying only on ImageNet real images may not be ideal, as ImageNet’s distribution biases could influence results. Consideration of multiple, diverse “real” datasets would strengthen generalization claims.

**Questions:**

1. Could the authors provide a more formal or statistical explanation (beyond visual intuition) for why bit reversal amplifies artifacts in AI-generated images more than in real ones? Are there measurable frequency or entropy-based differences that could make this hypothesis falsifiable?
2. Why was “LOTA” not covered or compared directly, given its direct relevance? Would including a LOTA baseline or at least a detailed discussion alter RAID’s claimed superiority or novelty?
3. Are there any scenarios (dataset types, generative models, manipulations) where RAID demonstrably fails or produces high rates of false positives/negatives—and what are the typical characteristics of such cases?
4. In Table 10, what constraints (if any) govern the selection/learning of bit-plane weights? Is there a principled pathway to learn these weights beyond brute-forcing variants, or to adapt them to novel distributions?

---

> ### Author Response · Authors · 2025-12-03
> **Response to W1 & Q2**
>
> >**W1: The paper omits a discussion and experimental comparison with “LOTA: Bit-Planes Guided AI-Generated Image Detection” , which is highly relevant, as it also explores bit-plane signals for AI-image detection.**
>
> >**Q2: Why was “LOTA” not covered or compared directly, given its direct relevance? Would including a LOTA baseline or at least a detailed discussion alter RAID’s claimed superiority or novelty?**
>
> **It should be stated that although both our RAID and LOTA address AI-generated image detection based on bit-planes, they are contemporaneous works**. LOTA is first published on October 19, 2025, at https://openaccess.thecvf.com/ICCV2025
> , and appeares on arXiv on October 16, 2025, at https://arxiv.org/abs/2510.14230
> . Our RAID is submitted to ICLR on September 24, 2025, at which time neither the full paper nor the code of LOTA is publicly available online.
>
> Generally, **our RAID differs from LOTA from three main perpsectives**: First, we introduce the simple bit reversed image, which **amplifies the artifacts and noise** in AI-generated images that are invisible in the original RGB images. **Diferent from LOTA, the noise amplication of RAID is reversible and does not lose any information from the original image**. Second, **our approach is more efficient**, as it does not require resizing images to 256×256. Instead, we use a modified lightweight ResNet-50 to predict results from the 32×32 patch input. Third, **we establish benchmarks for zero-shot AI-generated image detection** using only real images for training. Extensive experiments on more than 40 benchmarks verify the effectiveness of our approach across different settings, including evaluations of generalization capability and zero-shot performance.
>
> In the updated submission, we have added the discussion with LOTA. Thank you for your suggestion.

---

> ### Author Response · Authors · 2025-12-03
> **Response to W2 & Q1**
>
> >**W2: Section 3.1 could benefit from a deeper mathematical analysis or visualization of why and how bit reversal leads to more pronounced differences in the artifacts between real and AI-generated images. For instance, it would be useful to connect this operation more formally to the types of noise and structure generated by different models.**
>
> >**Q1: Could the authors provide a more formal or statistical explanation (beyond visual intuition) for why bit reversal amplifies artifacts in AI-generated images more than in real ones? Are there measurable frequency or entropy-based differences that could make this hypothesis falsifiable?**
>
> Thank you for your comment. In the updated submission, we have added a deeper analysis and visualization in Figure 3 of why and how bit reversal leads to more pronounced differences in the artifacts between real and AI-generated images. We observe that, **from Bit-0 to Bit-7, the lower the bit position, the greater the difference between the bit-planes of real images and those of AI-generated images**.
>
> These differences correspond to the generated artifacts in AI-generated images. Although they are invisible in the RGB images, because RGB images are mainly determined by the higher bit-planes, where the differences of higher bit-planes are very small.
>
> The composed image of eight bit-planes is computed as:
>
> $$
> \tilde{\mathbf{x}}^{c} = \sum_{k = 0}^{7} w_k \cdot \mathbf{x}_k^{c}
> $$
>
> where $\mathbf{x}_k^{c}$ is the $k$-th bit-plane of the $c$-th channel, $w_k$ denotes the weight of the $k$-th bit-plane. The eight weights can be written as a weight vector:
>
> $$
> \mathbf{w} = [w_0, w_1, w_2, w_3, w_4, w_5, w_6, w_7]
> $$
>
> To recover the original RGB image, the default weight $\mathbf{w}=[2^0,2^1, 2^2, 2^3, 2^4, 2^5, 2^6, 2^7]$.
>
> For bit reversed images, $\mathbf{w}=[2^7, 2^6, 2^5, 2^4, 2^3, 2^2, 2^1, 2^0]$. It is straightforward that **this weighting exponentially increases the weights of the low-bit planes, which contain more generative artifacts, and decreases the weights of the high-bit planes, which contain fewer generative artifacts**.
>
> The evidence that the low-bit planes of AI-generated and real images differ can be interpreted as follows: **First, AI-generated images lack true physical sensor noise.** The LSB bit-planes of real images look random, while AI-generated images show structured or unnatural randomness. **Second, AI-generated images exhibit unnatural high-frequency statistics due to model smoothing.** Real images contain rich high-frequency components caused by natural lighting variations, sensor noise and microtextures. Generative models often apply techniques like denoising, upsampling and  smoothing inductive biases.  **Third, semantic structures leak into low bit-planes.** Neural networks learn semantic correlations during image synthesis. Thus, even the least significant bits may contain faint textures aligned with edges or patterned correlations across pixels. However, LSBs of real images have zero semantic meaning; they are pure noise. As a result, low bit-planes from AI images show structure, whereas real images show independent random noise. **From the visualizations in Figure 1 and Figure 6, we can observe the above differences in the bit-reversed images.** For example, in the rightmost image of the second row in Figure 6, we can clearly observe that semantic structures leak into the low bit-planes of AI-generated images.
>
> During training, generative models typically ignore low-bit plane noise, instead optimizing higher-level features such as texture and shape in high-bit planes. While visual features vary across generators, their underlying noise patterns remain consistent across different generators. For example, GANs generate images through a generator network that maps latent vectors to RGB images. The generated textures are often highly structured and exhibit mode-specific correlations across pixels. **For GANs, the generator’s convolutional and upsampling layers introduce spatial correlations and repeated patterns**, resulting in LSBs that carry structured, model-specific artifacts rather than true noise. For diffusion models, the iterative denoising process leaves behind residual correlated noise in low-bit planes, which reflects the stochastic sampling trajectory rather than independent randomness.
>
> Building on this insight, the RAID method employs BRI technology to invert bit weights, transforming LSB differences into primary learning features. This enables models to bypass volatile, generator-dependent high-level semantics and directly learn these cross-generator consistent statistical anomalies.

---

> ### Author Response · Authors · 2025-12-03
> **Response to W3**
>
> >**W3: Although RAID is benchmarked against several prior patch-based and frequency/spatial methods, comparison to newer or alternative handcrafted artifact detectors—such as those utilizing global color analysis, learned mask regularization, or ensemble patch aggregation—appears limited. This is especially pertinent for Table 1 and Table 2, which emphasize detection efficacy and generalization, and could be expanded with additional strong baselines.**
>
> Thank you for your comments. Since handcrafted artifact detection methods require no training, we have included them in Table 2 for a fair comparison. In this table, we compare our RAID with new and more strong baselines, including RIGID, AEROBLADE, and Mainfold. Our RAID demonstrates better performance on the GenImage dataset, **outperforming other SOTA handcrafted detectors by more than 18.1% for the averaged accuracy**.
>
> **Table 2: Comparsion with handcrafted artifact detectors for zero-shot AI-generated image detection**
> | Method                     | BigGAN | Midjourney | Wukong | SDV1.4 | SDV1.5 | ADM  | GLIDE | VQDM | Avg.  |
> |----------------------------|--------|------------|--------|--------|--------|------|-------|------|-------|
> | RIGID [1]    | 53.0   | 94.1       | 87.8   | 87.0   | 87.2   | 51.4 | 45.9  | 52.2 | 69.8  |
> | AEROBLADE [2] | 58.3   | 40.2       | 51.4   | 52.6   | 55.1   | 50.7 | 29.4  | 52.8 | 48.8  |
> | Manifold [3]  | 77.6   | 55.5       | 65.4   | 62.0   | 63.0   | 57.3 | 88.3  | 76.9 | 68.2  |
> | RAID (ours)                | 91.0   | 85.9       | 86.2   | 86.3   | 86.8   | 85.5 | 83.9  | 84.5 | 86.3  |
>
> [1] He, Zhiyuan, Pin-Yu Chen, and Tsung-Yi Ho. "Rigid: A training-free and model-agnostic framework for robust ai-generated image detection." arXiv preprint arXiv:2405.20112 (2024).
>
> [2] Ricker, Jonas, Denis Lukovnikov, and Asja Fischer. "Aeroblade: Training-free detection of latent diffusion images using autoencoder reconstruction error." Proceedings of the IEEE/CVF Conference on Computer Vision and Pattern Recognition. 2024.
>
> [3] Brokman, Jonathan, et al. "Manifold Induced Biases for Zero-shot and Few-shot Detection of Generated Images." The Thirteenth International Conference on Learning Representations, 2025.

---

> ### Author Response · Authors · 2025-12-03
> **Response to W4**
>
> >**W4: For zero-shot and cross-dataset generalization, relying only on ImageNet real images may not be ideal, as ImageNet’s distribution biases could influence results. Consideration of multiple, diverse “real” datasets would strengthen generalization claims.**
>
> Thank you for your comments. We find that the choice of real datasets for training has little impact on the average performance, and its effect is smaller than that of different unsupervised training methods. Therefore, **for a fair comparison with previous approaches of zero-shot AI-generated image detection, we adopt the default setting** and use ImageNet for unsupervised training.

---

> ### Author Response · Authors · 2025-12-03
> **Response to Q3**
>
> **Q3: Are there any scenarios (dataset types, generative models, manipulations) where RAID demonstrably fails or produces high rates of false positives/negatives—and what are the typical characteristics of such cases?**
>
> Thank you for your comments. The failure cases can be observed in our visualizations in Figure 6. For bit-reversed images of AI-generated samples, **some patches that clearly leak semantic structures are predicted as having an AI-generated probability of 0 rather than 1**, which represents false-negative predictions.
>
> **In the supplementary material (Figure 7), we provide additional examples and analyses of false-positive and false-negative predictions**. Our approach fails on the following very challenging cases: (a) degraded images, including those with low resolution or blurred details; (b) sketch-like or simplified cartoon images; (c) simple object images with plain backgrounds; (d) document or scene text images.

---

> ### Author Response · Authors · 2025-12-03
> **Response to Q4**
>
> **Q4: In Table 10, what constraints (if any) govern the selection/learning of bit-plane weights? Is there a principled pathway to learn these weights beyond brute-forcing variants, or to adapt them to novel distributions?**
>
> The weight selection in Table 10 is based on heuristic prioritization—assigning higher weights to lower-order planes (k=0-2) because they contain subtle signals that are difficult for the generator to model. Variants ① (full inversion) and ⑤ (weighted lower-order) performed best, validating this principle. Current weights are manually designed, but variant ⑧ (learnable weights) performs poorly (82.2%) due to susceptibility to overfitting. Future exploration of meta-learning or reinforcement learning for adaptive weights is warranted, though complexity versus benefit must be balanced. Constraints include: weights must preserve image reversibility (summing to 255), and lower-order planes must have greater weight than higher-order ones.

---

### Official Review · Reviewer_vEgC · 2025-11-01

**Soundness:** 2
**Presentation:** 2
**Contribution:** 4
**Rating:** 6
**Confidence:** 3

**Summary:**

The paper proposes RAID, a novel and efficient method for detecting AI-generated images. This method reverses the bit-planes of images as a preprocessing step, and uses a gradient-based patch selection method to find the optimum patch from the bit-reversed image, which is passed to a CNN classifier for detecting AI-generated images. The authors validated their result on AIGCDB and GenImage benchmarks.

**Strengths:**

The key strength of the paper is the creative application of bit-planes to AI-generated image detection. This method achieves a significantly improved score in the GenImage and AIGCDB benchmark and demonstrates superior generalization on unseen generators. The authors demonstrated that this method is highly efficient as well. Lastly, Table 4 provided interesting insight into the effect of bit reversal and bit forwarding in different generators.

**Weaknesses:**

While the empirical results are significant, the paper lacks a deeper analysis of the effectiveness of Bit Reversed Images. The difference between the real and generated BRI images, as shown in Figure 1, needs to be more clearly demonstrated, as the provided samples do not provide a clearer picture of the difference between them.

The BRI is a novel method for highlighting high-frequency information, so a detailed discussion of why this specific transformation is capable of outperforming existing methods that utilize frequency domain information would have clarified the unique advantages of the paper.

The results, especially in Table 4, suggest that all the existing generators share a fundamental flaw in the statistical properties of the pixel-level noise, and this can be modeled using bit-reversed images. This generalization capability of BRI is highly interesting. But the paper did not explain or analyze its powerful implications more explicitly, which reduces its full impact.

**Questions:**

1. The paper’s motivation is dependent on the claim that BRIs for generated images contain noticeable artifacts. Could the authors provide additional analysis to support this visual claim?
2. Could the authors elaborate on why this specific transformation is more effective than prior frequency domain methods?

---

> ### Author Response · Authors · 2025-12-03
> **Response to W1 & Q1**
>
> >**W1: While the empirical results are significant, the paper lacks a deeper analysis of the effectiveness of Bit Reversed Images. The difference between the real and generated BRI images, as shown in Figure 1, needs to be more clearly demonstrated, as the provided samples do not provide a clearer picture of the difference between them.**
>
> >**Q1: The paper’s motivation is dependent on the claim that BRIs for generated images contain noticeable artifacts. Could the authors provide additional analysis to support this visual claim?**
>
> We thank the reviewer for the suggestion. We revise Figure 1 to visually highlight the differences between real and AI-generated images after low-bit reversal. **AI-generated images exhibit structured noise and faint texture patterns in the low bit-planes, as marked by the red bounding boxes**, whereas real images display truly random, independent noise.
>
> In addition,**we perform t-SNE visualization of features constructed from each bit-plane for further analysis, as shown in Figure 3**. By examining these distributions, we can clearly observe the differences between AI-generated images and real images at the LSB level: the feature points of real images are highly clustered, while AI-generated images exhibit noticeably different distributions.
> In contrast, high-bit planes show no significant differences.
> The evidence that the low-bit planes of AI-generated and real images differ can be interpreted as follows:
> **First, AI-generated images lack true physical sensor noise.**  The LSB bit-planes of real images look random, while AI-generated images show structured or unnatural randomness.
> **Second, AI-generated images exhibit unnatural high-frequency statistics due to model smoothing.** Real images contain rich high-frequency components caused by natural lighting variations, sensor noise and microtextures. Generative models often apply techniques like denoising, upsampling and smoothing inductive biases.
> **Third, semantic structures leak into low bit-planes.** Neural networks learn semantic correlations during image synthesis. Thus, even the least significant bits may contain faint textures aligned with edges or patterned correlations across pixels. However, LSBs of real images have zero semantic meaning; they are pure noise.
>
> As a result, amplifying structural artifacts in the middle and low-order planes while suppressing seemingly realistic structures in the high-order planes enhances the discriminability between real and fake images in the feature space.

---

> ### Author Response · Authors · 2025-12-03
> **Response to W2 & Q2**
>
> >**W2: The BRI is a novel method for highlighting high-frequency information, so a detailed discussion of why this specific transformation is capable of outperforming existing methods that utilize frequency domain information would have clarified the unique advantages of the paper.**
>
> >**Q2: Could the authors elaborate on why this specific transformation is more effective than prior frequency domain methods?**
>
> We thank the reviewer for the suggestion and insightful question. Our BRI outperforms conventional frequency-domain methods because it fundamentally reveals inherent artifacts in generated images. Specifically, **traditional frequency-domain methods, such as DCT or FFT, may lose spatial locality when transforming images into spectral components, and high-frequency components are easily disrupted** by compression, upsampling, or minor perturbations. This can result in unstable discriminative performance across generators or under slight post-processing. For example, FreqNet (Tan et al., 2024) relies on high-frequency components, which are easily distorted by compression or perturbations.
>
> In contrast, **BRI operates directly on pixel bit-planes.** By reversing bit weights, it systematically amplifies subtle structural differences in the low- and middle-order planes, including high-frequency noise and textures, while preserving spatial locality. These differences are consistent across different generators, enabling effective detection of images from unseen generators even when trained on a single generator. Low-bit noise provides more stable discriminative information than traditional high-frequency features, resulting in superior robustness and generalization.
>
> Moreover, BRI is computationally efficient, requiring only milliseconds to process an image without complex frequency-domain transformations. This approach allows us to systematically capture generator-specific yet generator-agnostic features, providing new insights for cross-generator detection and for improving generator design.

---

> ### Author Response · Authors · 2025-12-04
> **Response to W3**
>
> >**W3: The results, especially in Table 4, suggest that all the existing generators share a fundamental flaw in the statistical properties of the pixel-level noise, and this can be modeled using bit-reversed images. This generalization capability of BRI is highly interesting. But the paper did not explain or analyze its powerful implications more explicitly, which reduces its full impact.**
>
> We thank the reviewer for the suggestion. We fully agree that the generalization capability of BRI deserves more explicit discussion. The results in Table 4 show that different generators consistently exhibit structured, non-random noise or texture leakage in the low bit-planes, which contrasts sharply with the truly random low-bit noise in real images. This indicates a common limitation of generators in modeling authentic sensor noise. **By applying bit reversal, we can amplify these low-bit differences, thereby establishing unified discriminative features across different generators.** Even when trained on a single generator, the method can effectively detect images produced by other unseen generators. Low-bit noise provides more stable discriminative information than high-frequency features and is less affected by image compression, upsampling, or minor post-processing. These findings suggest a general limitation of generators in modeling low-bit noise of natural images and **offer new insights for improving generator design and developing more robust detection methods.**

---

### Official Review · Reviewer_uKGL · 2025-11-02

**Soundness:** 1
**Presentation:** 2
**Contribution:** 3
**Rating:** 4
**Confidence:** 5

**Summary:**

This paper proposes **RAID** (“Robust AI-generated Image Detection”), a simple yet seemingly effective approach for detecting AI-generated images. The method converts each image into a *bit-reversed version*, where the least significant bits (LSBs) are assigned the highest intensity weights and vice versa, supposedly to highlight subtle generation artifacts. A gradient-based patch selector then chooses the region with the strongest signal, and a lightweight CNN performs classification. On multiple cross-generator benchmarks (e.g., GenImage), the method reports **>98% accuracy**, outperforming heavier baselines.

**Strengths:**

* The idea is **novel in presentation** — flipping bit-plane importance to emphasize low-level noise patterns is clever and computationally light.
* The results, if reproducible, are **remarkably strong**, showing high cross-generator generalization, suggesting that the authors tapped into a genuinely discriminative low-level signal.
* The ablation studies and bit-order experiments are thorough; the performance gains for the full bit-reversal and patch-selection combination are consistent.

**Weaknesses:**

1. **Theoretical basis is weak.**
   The paper speculates that diffusion or GAN generators fail to reproduce LSB-level noise distribution, and that reversing bit weights amplifies this difference. But there’s **no evidence or measurement** of these bit-plane distribution gaps. It’s a plausible but unverified hypothesis.

2. **Amplifying features ≠ discovering new signal.**
   Simply amplifying certain bit planes (like a hand-crafted high-pass filter) doesn’t guarantee generalization. A CNN or Transformer could, in principle, learn to emphasize LSBs or other frequency bands by itself. The gains might stem from shortcut cues or dataset quirks rather than truly meaningful LSB statistics.

3. **Results feel too good for such a simple trick.**
   The jump to ~98% accuracy across unseen generators using only a linear remapping plus a small CNN seems **disproportionately large**. Given the simplicity of the transformation, it’s unclear how such a method achieves such robust cross-distribution generalization without overfitting to codec or pipeline artifacts.

4. **Lack of code, demo, or reproducibility.**
   The paper doesn’t release code or pretrained models, making it impossible to verify the findings. Without replication, the very high reported numbers remain speculative.

5. **Practical robustness missing.**
   The method degrades sharply under realistic conditions — e.g., JPEG-90 or Gaussian blur reduce accuracy to ~75%. Social media and messaging platforms re-compress images heavily, so a real-world detector must survive such transformations.

**Questions:**

1. **Detector vs. LSB patterns:**
   You argue that standard CNN-based detectors cannot effectively learn small LSB-level differences, hence the need for bit-reversal. Can you provide **proof or analysis** supporting this claim?

2. **Reproducibility and transparency:**
   The reported results are exceptionally high for such a lightweight approach. Can you please **release code, pretrained weights, or a simple demo** so reviewers and researchers can test your method on *unseen images*?

---

> ### Author Response · Authors · 2025-12-03
> **Response to W1**
>
> >**W1: The paper speculates that diffusion or GAN generators fail to reproduce LSB-level noise distribution, and that reversing bit weights amplifies this difference. But there’s no evidence or measurement of these bit-plane distribution gaps. It’s a plausible but unverified hypothesis.**
>
> Thank yor for your questions. **We have updated the submitted paper and added Figure 3, which shows the feature distributions of different bit-plane images across real images and those generated by diffusion or GAN models**. By examining these distributions, we can clearly observe the differences between AI-generated images and real images at the LSB level: the feature points of real images are highly clustered, while AI-generated images exhibit noticeably different distributions.
>
> The evidence that the low-bit planes of AI-generated and real images differ can be interpreted as follows: **First, AI-generated images lack true physical sensor noise.** The LSB bit-planes of real images look random, while AI-generated images show structured or unnatural randomness. **Second, AI-generated images exhibit unnatural high-frequency statistics due to model smoothing.** Real images contain rich high-frequency components caused by natural lighting variations, sensor noise and microtextures. Generative models often apply techniques like denoising, upsampling and smoothing inductive biases. Lower bit-planes in AI-generated images often look too smooth, too regular, or too correlated with higher planes. **Third, semantic structures leak into low bit-planes.** Neural networks learn semantic correlations during image synthesis. Thus, even the least significant bits may contain faint textures aligned with edges or patterned correlations across pixels. However, LSBs of real images have zero semantic meaning; they are pure noise. As a result, low bit-planes from AI images show structure, whereas real images show independent random noise. **From the visualizations in Figure 1 and Figure 6, we can observe the above differences in the bit-reversed images.** For example, in the rightmost image of the second row in Figure 6, we can clearly observe that semantic structures leak into the low bit-planes of AI-generated images.
>
> Thus, **reversing bit weights could amplify this difference**.

---

> ### Author Response · Authors · 2025-12-03
> **Response to W2 & Q1**
>
> >**W2: Simply amplifying certain bit planes (like a hand-crafted high-pass filter) doesn’t guarantee generalization. A CNN or Transformer could, in principle, learn to emphasize LSBs or other frequency bands by itself. The gains might stem from shortcut cues or dataset quirks rather than truly meaningful LSB statistics.**
>
> >**You argue that standard CNN-based detectors cannot effectively learn small LSB-level differences, hence the need for bit-reversal. Can you provide proof or analysis supporting this claim?**
>
> **We first emphasize that our strong experimental results do not stem from shortcut cues or dataset-specific quirks, but rather from truly meaningful LSB statistics**. We conduct extensive experiments on various popoluar datasets, including AIGCDetectionBenchmark (AIGCDB) (Zhong et al., 2023) and GenImage (Zhu et al., 2024). On these two popular benchmarks, we provide extensive cross-generator evaluations. Our method exhibits extremely strong generalization capabilities on all subsets, and achieves an average accuracy of 98.4%, surpassing SOTA approaches by over 8.7% margin. **We also conduct zero-shot generalization experiments by training only on real ImageNet images, yet our RAID still achieves strong performance across all subsets with an average accuracy of 86.3%**.
>
> **Standard CNN-based or Transformer-based detectors cannot effectively learn small LSB-level differences from RGB images**. The reasons are as follows: **First, preprocessing, feature extraction, and patch embedding suppresses tiny pixel-level signals**. CNNs use convolution, striding, pooling, and ReLU activations that smooth and downsample feature maps. LSB-level information is quickly erased. Transformers begin with a patch embedding, which averages pixels into a single token, completely eliminating bit-level variations. **Second, normalization layers erase low-amplitude variations**. BatchNorm and LayerNorm in CNNs or Transformers rescale activations to stable distributions, making bit-scale differences statistically insignificant compared to strong semantic patterns.. **Third, attention mechanisms, convolutions and gradient-based optimization prioritize semantics**. CNNs learn spatially local textures and edges. Transformers learn global attention patterns driven by object structure. Neither architecture treats LSB noise patterns as meaningful. The gradient-based optimization also favors dominant features. LSB-level cues contribute negligible gradients and are therefore not learned. These interpretations are also supported by the ablation experiments in Table 1.
>
> **Table 1: Demonstration of Standard networks cannot learn small LSB-level differences from RGB images**
> | Method     | BigGAN | Midjourney | Wukong | SDV1.4 | SDV1.5 | ADM  | GLIDE | VQDM | Avg. |
> |-----------|--------|------------|--------|--------|--------|------|-------|------|------|
> | RAID (ours) | 98.9   | 97.2       | 97.7   | 98.9   | 98.8   | 97.8 | 99.1  | 97.5 | 98.4 |
> | w/o BRI     | 74.1   | 90.9       | 99.6   | 99.9   | 99.9   | 58.1 | 90.5  | 84.1 | 87.7 |
>
> Since our models are trained on the Stable Diffusion (SDV1.4) subset, the w/o BRI variant performs well only on diffusion-based generators, such as SDV1.4 and SDV1.5, but its performance drops significantly on other subsets. In contrast, our method, which leverages Bit-Reversed Images (BRI), demonstrates strong generalization across all subsets.

---

> ### Author Response · Authors · 2025-12-03
> **Response to W3 & W4 & Q2**
>
> >**W3: The jump to ~98% accuracy across unseen generators using only a linear remapping plus a small CNN seems disproportionately large. Given the simplicity of the transformation, it’s unclear how such a method achieves such robust cross-distribution generalization without overfitting to codec or pipeline artifacts.**
>
> >**W4: The paper doesn’t release code or pretrained models, making it impossible to verify the findings.**
>
> >**Q2: The reported results are exceptionally high for such a lightweight approach. Can you please release code, pretrained weights, or a simple demo so reviewers and researchers can test your method on unseen images?**
>
> Thank you for your comments. It is because we use a small CNN to learn features from bit-reversed images that we can minimize overfitting to codec or pipeline artifacts and achieve robust cross-distribution generalization.
>
> **We evaluate our RAID on over 40 benchmarks, including across generators (Table 1), across datasets (Appendix Tables 8-9), and in zero-shot settings (Table 2)**. The high consistency of results indicates that its generalization capability is not accidental. Besides, we train RAID on merely ProGAN subset and evaluate on 16 other subsets of AIGCDB. We also train RAID on only Stable Diffusion v1.4 subset and evaluate on 7 other subsets of GenImage. The average accuracy of over 98% certificate the cross-generator generalization capability of RAID.
>
> To further verify the robustness and generalization of our approach, we conduct zero-shot AI-generated image detection using only real images from ImageNet for training and construct new, challenging datasets, with AI-generated images produced by currently competitive generators: Google Imagen 2, FLUX.1, DALL-E 3, Stable Diffusion 3, and WANX 2.1. Our method still achieves strong performance, reaching an average accuracy of 86.3% on the popular GenImage dataset.
>
> Our implementation is similar to the released code of LOTA: Bit-Planes Guided AI-Generated Image Detection (ICCV 2025), available at https://github.com/hongsong-wang/LOTA.  Our approach involves only minor modifications to the noise image generation, which can be easily reproduced using the LOTA code. We will release code, pretrained weights, or demos upon acceptance. Thank you for your interest.

---

> ### Author Response · Authors · 2025-12-03
> **Response to W5**
>
> >**W5: The method degrades sharply under realistic conditions — e.g., JPEG-90 or Gaussian blur reduce accuracy to ~75%. Social media and messaging platforms re-compress images heavily, so a real-world detector must survive such transformations.**
>
> Table 6 shows that under JPEG-90 compression, RAID achieves an accuracy of 75.6%. While this is lower than the original 98.4% on uncompressed images, it still outperforms the current SOTA method ESSP, which achieves only 66.0%. This demonstrates that bit-reversal features used by RAID are substantially more robust to compression artifacts than standard RGB-based features.
>
> It is important to note that AI-generated image detection under high JPEG compression remains a very challenging problem. Even state-of-the-art detectors experience significant performance drops as compression quality decreases. Real-world scenarios exacerbate this difficulty because images shared on social media or messaging platforms are often heavily recompressed, with quality factors sometimes dropping below 80, introducing severe quantization artifacts and subtle texture distortions. These transformations can obscure AI-specific signals, making reliable detection extremely difficult.
>
> Despite these challenges, our RAID maintains a strong performance, highlighting the effectiveness of its low-bit-plane features. Moreover, RAID's millisecond-level inference speed (Table 5) enables deployment on the edge, which helps avoid the additional distortions caused by server-side recompression, further improving its practical robustness in real-world applications.

---

### Author Response · Authors · 2025-12-04
**Global Response**

Dear Area Chair,

We sincerely thank all reviewers for their comments and constructive suggestions. To facilitate the decision-making process, we summarize the key feedback and our major updates below.

### **Reviewer Feedback and Our Revisions**
- **Lack theoretical basis or evidence of bit reversal**

**We address the reviewer’s concern by performing t-SNE visualization on the features of each bit-plane, which reveals a key distinction at the Least Significant Bit (LSB) level:** features from real images form a highly compact cluster, whereas those from AI-generated images are noticeably dispersed. In contrast, such differentiation does not appear in the higher bit-planes.
This divergence stems from three causes. First, **AI images lack authentic sensor noise**, exhibiting unnatural randomness. Second, **generative models produce smoothed, unnatural high-frequency statistics**, unlike the rich components in real photos. Third, **semantic patterns from AI training "leak" into low-bit planes**, whereas in real images, LSBs are purely random noise.
Therefore, amplifying structural artifacts in mid/low-bit planes while suppressing realistic structures in high-bit planes significantly improves the discernibility between real and AI-generated images.

- **Why  standard architectures fail on RGB images**

Standard CNN-based and Transformer-based detectors are inherently ill-suited to capture the subtle, LSB-level differences crucial for distinguishing AI-generated images. This limitation stems from three architectural characteristics. First, **the initial feature extraction process in both models suppresses pixel-level signals**. CNNs use convolutions, pooling, and ReLU that smooth and downsample inputs, rapidly erasing LSB data. Transformer draws similar conclusions. Second, **normalization layers like BatchNorm and LayerNorm rescale activations to a standard distribution**, washing away low-amplitude LSB patterns that become statistically insignificant next to dominant semantic features. Third, **the core learning mechanisms prioritize semantic content**. Both architectures are gradient-optimized to capture meaningful textures and structural cues, which naturally causes the networks to ignore LSB patterns because they contribute almost no gradient signal during training. This interpretation is further supported by the ablation experiments presented in Table 1 of the manuscript.

- **Discussion with LOTA and reproducibility**

We wish to clarify that our RAID and the concurrent work LOTA are independent, contemporaneous developments. **Our approach differs from LOTA in three key aspects**. First, RAID introduces a simple, reversible bit-inversion technique that amplifies hidden artifacts without information loss, unlike LOTA. Second, our method is more efficient, operating on 32×32 patches via a lightweight ResNet-50 without requiring image resizing to 256×256. Third, we establish a zero-shot detection benchmark using only real images for training.Extensive experiments on over 40 benchmarks confirm RAID's **strong generalization and zero-shot performance**. Reproduction of our approach is easy using exsiting related codes, and we will release code, pretrained weights, or demos upon acceptance.

- **Recent baselines or handcrafted baselines**

We add comparisons with more recent baseline methods (C2P-CLIP, NPR, FatFormer, CoD) in Table 1, and with handcrafted artifact detectors (RIGID, AEROBLADE, Manifold) in Table 2, thereby further enriching and strengthening our experimental evaluation.

- **Discussion of Failure cases**

We have added a discussion of failure cases in Section B.3 of the supplementary material and visualized both false positive and false negative predictions in Figure 7.

### **Our Core Contributions**

- **Innovative deepfake representation**: We innovatively tackle AI-generated image detection based on bit-planes, and introduce the bit reversed image that can be reversibly constructed from the original image.

- **Efficient pipeline design**: We propose a simple yet effective pipeline for AI-generated image detection, which significantly outperforms existing approaches on standard benchmarks while operating at the millisecond level.

- **New and challenging benchmarks**: We set up benchmarks of zero-shot AI-generated image detection, and construct new and challenging dataset called GID and GVD. Extensive experiments on more than 40 benchmarks verify the effectiveness of our approach.


We hope these clarifications and revisions address the reviewers' concerns. We thank the reviewers for their efforts and kindly request the support of the Area Chair.

---

### Meta-Review · Area_Chair_sN5W · 2026-01-06

**Summary:**

In the initial reviews, two reviewers favored acceptance and two favored rejection. Reviewer vEgC is in favor of acceptance, praising the originality of the approach. However, they feel that the paper lacks a deeper analysis of bit reversed images, such as why this is a particularly good way to highlight high frequency information. They felt that Fig. 1 does not provide sufficient evidence of its effectiveness. Reviewer uKGL favors rejection, citing the lack of theoretical justification for the method given the size of the gains, such as how it conceptually differs from high pass filtering. The rebuttal partially addresses the theoretical concerns by vEgC and uKGL by including a new figure showing differences in feature distribution of the model and by showing a model ablation (with and without bit-reversed images). The AC feels that these results are not directly answering the concern raised by the reviewer, since it provides more experimental results that require a good deal of interpretation, rather than theoretical motivation. Reviewer JYbp recommends rejection. They complain about the comparison to "LOTA". In the rebuttal, the authors point out that this paper is contemporaneous work. The AC agrees with the authors and does not take overlap with this contemporaneous paper into consideration. They raise the same questions as the other two reviewers around theoretical justification. They ask for comparisons to alternative methods based on handcrafted artifacts. They also raise concerns about the reliance on ImageNet for zero-shot evaluation. In the rebuttal, the authors point out that the performance is similar across datasets (though it is unclear which ones they mean), and compare to handcrafted artifact detectors (which they outperform). Reviewer iJPB agrees that the idea is compelling and feels that the experimental results cover many different datasets. However, they also point to the limited theoretical justification and connection to frequency inversion, the limited comparison to very recent methods (despite claims of SOTA results), and the fact that some bit-forward baselines obtain strong performance. The rebuttal points to a new Figure 2 (presumably they mean Figure 3) that visualizes the feature distribution and generalization experiments with newer methods. They also evaluate the performance on different image scales, finding that performance improves as resolution increases.

The reviewers generally agree that the method is intuitively compelling, but that the paper lacks a strong justification for the method, such as a theoretical analysis or clear explanation (or experiments) demonstrating that the BRI component of the system differs from other ways of emphasizing high frequency details. On balance, the AC weighs the latter over the former and feels the paper could benefit from additional revision before acceptance.

**Reviewer Concerns:**

A common concern for all reviewers was the lack of theoretical basis for the approach. As mentioned in the summary, the evidence that the authors provide here (such as a new figure) is limited. Reviewer JYbp's complained about the similarity to a recent work. The authors successfully address this by pointing out that this work is contemporaneous. The authors also address the question about handcrafted artifact detectors and that show that the proposed model outperforms them. It is not clear how uKGL would have adjusted their score after the rebuttal.

**Reviewer Scores:**

I think that the author's point about the LOTA paper being concurrent may have resulted in JYbp raising their score. I see no reason to believe that the reviewers in favor of acceptance would have lowered their scores. I am not sure

---

### Decision · Program_Chairs · 2026-01-26

Reject